# Committed sea-level rise under the Paris Agreement and the legacy of delayed mitigation action

Matthias Mengel [1], Alexander Nauels [2], Joeri Rogelj [3,4,5] & Carl-Friedrich Schleussner [1,6,7]

Sea-level rise is a major consequence of climate change that will continue long after emissions of greenhouse gases have stopped. The 2015 Paris Agreement aims at reducing climate-related risks by reducing greenhouse gas emissions to net zero and limiting global-mean temperature increase. Here we quantify the effect of these constraints on global sea-level rise until 2300, including Antarctic ice-sheet instabilities. We estimate median sea-level rise between 0.7 and 1.2 m, if net-zero greenhouse gas emissions are sustained until 2300, varying with the pathway of emissions during this century. Temperature stabilization below 2 °C is insufficient to hold median sea-level rise until 2300 below 1.5 m. We find that each 5-year delay in near-term peaking of $CO_2$ emissions increases median year 2300 sea-level rise estimates by ca. 0.2 m, and extreme sea-level rise estimates at the 95th percentile by up to 1 m. Our results underline the importance of near-term mitigation action for limiting long-term sea-level rise risks.

[1] Potsdam Institute for Climate Impact Research (PIK), Member of the Leibniz Association, P.O. Box 60 12 03D-14412 Potsdam, Germany. [2] Australian-German College of Climate and Energy Transitions, The University of Melbourne, Parkville, VIC 3010, Australia. [3] ENE Program, International Institute for Applied Systems Analysis (IIASA), Schlossplatz 1, Laxenburg, A-2361, Austria. [4] Institute for Atmospheric and Climate Science, ETH Zurich, Universitätstrasse 16, Zurich, 8006, Switzerland. [5] School of Geography and the Environment, Oxford University, South Parks Road, OX1 3QY Oxford, UK. [6] Climate Analytics, Ritterstr. 3, 10969 Berlin, Germany. [7] Integrative Research Institute on Transformations of Human-Environment Systems (IRI THESys), Humboldt-Universität zu Berlin, 10969 Berlin, Germany. Correspondence and requests for materials should be addressed to M.M. (email: matthias.mengel@pik-potsdam.de)

At the 21st UNFCCC climate conference in Paris, countries renewed their global commitment to combat climate change and its impacts. The Paris Agreement[1] sets a temperature goal of holding the increase in the global-mean temperature well below 2 °C above pre-industrial levels and pursuing efforts to limit it to 1.5 °C above pre-industrial levels. To accomplish this, the agreement aims at peaking global greenhouse gas (GHG) emissions as soon as possible and achieving 'a balance between anthropogenic emissions by sources and removals by sinks of greenhouse gases in the second half of the 21st century'. This balance can be interpreted as achieving net-zero GHG emissions between 2050 and 2100 (ref.[2, 3]).

Sea-level rise is one of the major consequences of anthropogenic climate change[4–6] and the effects of sea-level rise in its combination with storm surges and land subsidence can already be observed today[7]. Global sea-level rise consists of the sum of several components in response to a common forcing, and shows a slow and delayed response to today's atmospheric warming and GHG emissions. Thermal expansion of ocean water, the retreat of mountain glaciers and ice caps, and the mass loss of the Greenland and Antarctic ice sheets are the main drivers of sea-level rise linked to climate change. These contributors respond in different ways to a warmer climate, and all respond on timescales that range from centuries to millennia[8]. GHG emissions of today and of the near future hence commit the Earth system to a sea-level rise legacy, which will only fully unfold in the centuries to come.

Process-resolving[4] and semi-empirical[9, 10] assessments of sea-level rise under climate change predominantly utilize the RCP scenarios[11] and are often limited to the 21st century. Probabilistic estimates have been published for the 22nd century[12] and the importance of climate policy was highlighted for sea-level rise during this century[13] and until 2300 (ref. [14]). A recent 10,000-year perspective of 21st century climate policy and sea-level rise[8] extends earlier assessments of multi-millennial sea-level commitment[15]. What has not been examined so far are the sea-level rise implications of achieving the Paris Agreement's temperature and mitigation goals over the coming century and beyond.

We here explore the sea-level legacy until 2300 within the constraints of the Paris Agreement with a reduced-complexity carbon cycle and climate model ensemble[16, 17] and a component-based simple sea-level model[18]. The sea-level model accounts for new evidence showing a higher sensitivity of the Antarctic ice sheet to global climate change[19]. We calibrate the Greenland component to newer observations that include the recent years of high-mass loss[20, 21]. We apply the methodology of ref. [18] without changes for thermal expansion and glaciers, which yield estimates similar to the IPCC AR5 for 21st century sea-level rise[4]. We find that sea level continues to rise in almost all cases throughout 2300, with sea-level stabilization possible but not probable under declining global-mean temperatures. In our scenarios, near-term emissions reductions strongly affect long-term sea-level rise as post-2050 emissions are constrained by the Paris Agreement.

## Results

**Paris Agreement scenarios**. We investigate scenarios that achieve the net-zero GHG emission goal of the Paris Agreement and hold global-mean temperature rise at various levels below 2 °C (Fig. 1d). We also explore scenarios that only achieve net-zero $CO_2$ emissions, while still stabilizing temperature rise below 2 °C (Fig. 1a). Throughout the manuscript, we discuss the results for these two subsets of our scenario ensemble. Our stylized emissions scenarios vary $CO_2$ emissions of fossil-fuel use and industry only, and are otherwise equal to RCP2.6. The reduction rates after peak emissions (0.3, 0.5, and 0.7 $GtC\,yr^{-2}$) are set to span the range between the minimal rate for reaching the Paris

temperature goal with very early peak emissions and the maximum rate assessed in the literature[22]. See Methods section for a detailed description.

**Projected global sea-level rise**. The combination of a reduced-complexity, uncertainty-propagating model of global-mean temperature with a component-based simple sea-level model allows us to assess the implications of different emission pathways on sea-level rise until 2300 (Methods section). Median sea-level rise reaches 116–164 cm in 2300 under temperature stabilization (net-zero $CO_2$) scenarios (Fig. 1c, Table 1a) and 73–123 cm under net-zero GHG scenarios (Fig. 1f, Table 1b; all absolute sea-level rise projections are expressed relative to 2000 levels). The combined uncertainty of the climate response to emissions and the sea-level response to climate warming is asymmetric (Fig. 1c, f) and dominated by the high sensitivity of the Antarctic ice sheet under high warming (Figs. 2d and 3d). We find that a risk of sea-level rise of up to 5 m (or 3 m) is within the 90% uncertainty range for three net-zero $CO_2$ (respectively, net-zero GHG) scenarios at the higher end of our ensemble (Table 1).

Although median sea-level rises continuously under all emission pathways through at least year 2300, this is not the case for the rate at which sea level changes. The rate of median sea-level rise starts to slow down shortly after emissions peak and continues to decline thereafter (Supplementary Fig. 2). Also under our pathways with the fastest and earliest emission reductions, median sea-level rise has not yet stopped by 2300 despite falling global-mean temperatures (Fig. 1e and Supplementary Fig. 2b). Median rates at the end of the 22nd century drop to 0.06–0.17 $cm\,yr^{-1}$ for net-zero GHG scenarios and to 0.33–0.49 $cm\,yr^{-1}$ for net-zero $CO_2$ scenarios (minimum and maximum within scenario ensemble, Table 2). The 16.6th percentile estimate for net-zero GHG scenarios indicates that sea level stabilization is possible under declining global-mean temperatures.

**Responses of individual sea-level components**. The four individual sea-level rise contributions for net-zero $CO_2$ (temperature stabilization) scenarios are displayed in Fig. 2. The median contributions have the same order of magnitude. Due to the high sensitivity to global temperature change of the Antarctic component[19], the low-probability high-end estimate is dominated by the ice sheets. A 3 m contribution from Antarctica cannot be ruled out for our warmest scenario, which achieves temperature stabilization through net-zero $CO_2$, peaks emissions in year 2035 and thereafter follows a reduction rate of 0.7 $GtC\,yr^{-2}$. This is >2 m above the most stringent temperature stabilization scenario in our set with peak year 2020 and subsequent reduction rate of 0.7 $GtC\,yr^{-2}$ (Supplementary Data 4). A negative Antarctic contribution is not probable but possible under temperature stabilization. In contrast, thermal expansion, mountain glaciers and the Greenland ice sheet continue to add to sea-level rise under temperature stabilization (see also Supplementary Fig. 3). Under the Paris Agreement's net-zero GHG constraint and the implied declining global-mean temperatures, all contributions except Greenland can stabilize during the 22nd century (Fig. 3 and Supplementary Fig. 4). Thermal expansion, mountain glaciers and the Antarctic ice sheet can contribute negatively toward the end of the 22nd century in our model, but only the Antarctic contribution can be significantly negative. See Methods section for more details on the responses of the individual components, including a comparison to results from more comprehensive models. Percentile estimates for each sea-level component

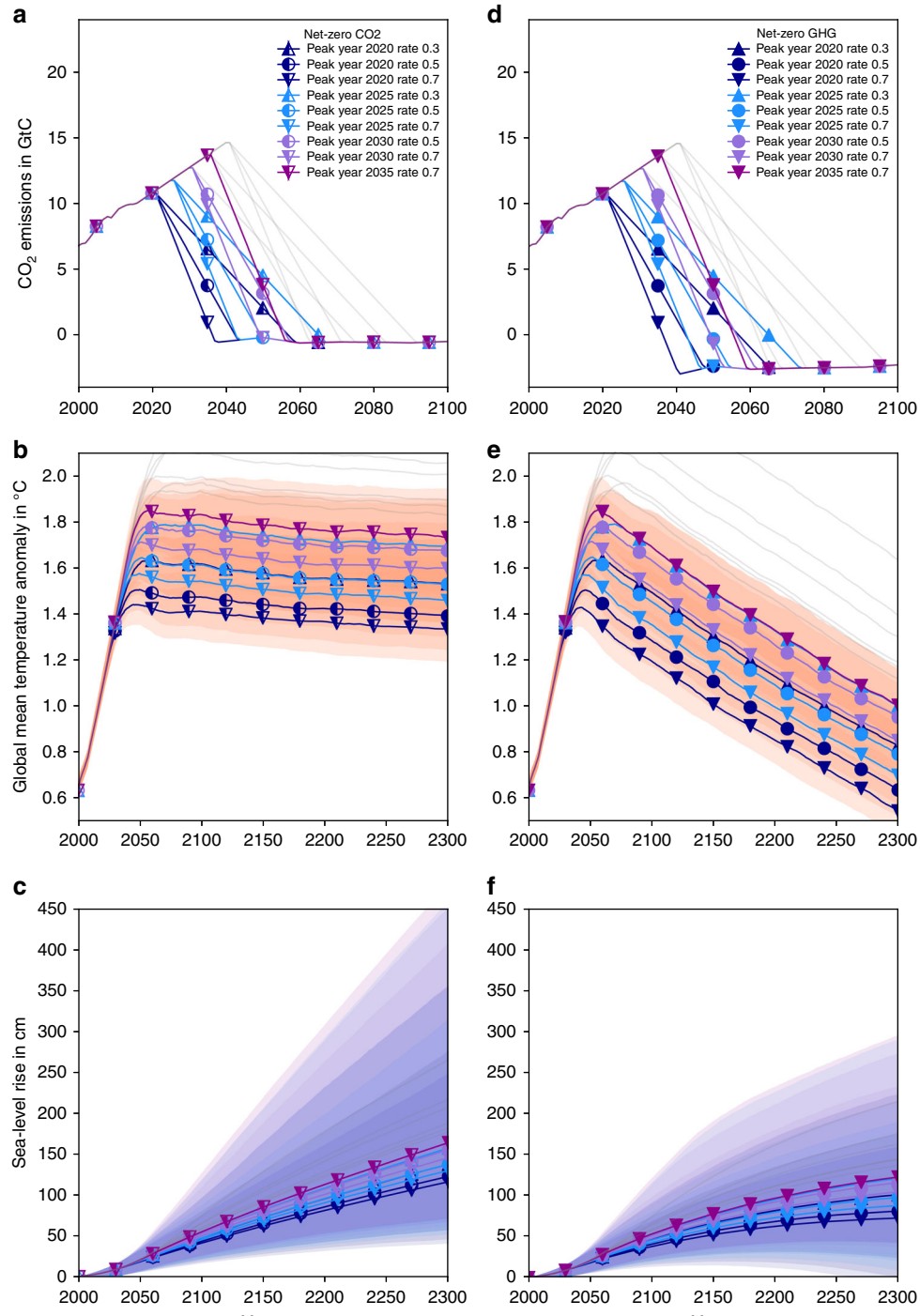

**Fig. 1** $CO_2$ emissions and respective global-mean temperature and sea-level responses. Emission scenarios based on RCP2.6 with $CO_2$ emissions from fossil-fuel use and industry linearly continued with the present day rate until peak year. $CO_2$ emissions decline by 0.3, 0.5, and 0.7 GtC $yr^{-2}$ thereafter until net-zero $CO_2$ **a** or net-zero greenhouse gas emissions **d** are reached. Scenarios that do not hold warming to below 2 °C with at least 66% chance are masked gray. **b**, **e** Global-mean temperature responses to emissions scenarios **a**, **d** in °C above pre-industrial levels. **c**, **f** Global-mean sea-level rise relative to the year 2000. Shading refers to the central 66th percentile range per scenario in **b**, **e** and to the central 90th percentile range in **c**, **f**

response to each net-zero $CO_2$ and net-zero GHG scenario are provided as Supplementary Data 4.

**Linking sea-level rise and near-term climate action.** Our scenario ensemble allows us to assess how 2300 sea-level rise changes as a function of our scenario parameters. All other parameters being equal, median sea-level rise in 2300 is about 40 cm higher in

scenarios that only stabilize temperature rise by keeping global $CO_2$ emissions at net-zero levels, compared to scenarios that additionally reach net-zero GHG emissions and thus peak and decline global-mean temperatures (Fig. 4a). The spread within each of these two scenario groups is of similar magnitude. Since emission levels toward the end of the century are constrained by the Paris Agreement's net-zero goal, near-term emissions (for example in the year 2030) become a predictor for 2300 sea-level

**Table 1 Total climate-driven sea-level rise for (a) net-zero CO$_2$ (temperature stabilization) and (b) net-zero GHG scenarios in 2300 relative to the year 2000**

| Percentile | 5.0 | 16.66 | 50.0 | 83.33 | 95.0 |
|---|---|---|---|---|---|
| **(a)** | | | | | |
| Peak year 2020 rate 0.3 | 56.67 | 84.41 | 137.27 | 219.66 | 355.43 |
| Peak year 2020 rate 0.5 | 45.66 | 74.36 | 122.5 | 189.87 | 275.81 |
| Peak year 2020 rate 0.7 | 40.37 | 70.24 | 115.88 | 178.43 | 247.87 |
| Peak year 2025 rate 0.3 | 67.47 | 96.95 | 156.98 | 268.83 | 458.3 |
| Peak year 2025 rate 0.5 | 56.69 | 84.41 | 137.29 | 219.65 | 354.76 |
| Peak year 2025 rate 0.7 | 51.59 | 79.36 | 129.64 | 203.87 | 313.75 |
| Peak year 2030 rate 0.5 | 66.54 | 95.83 | 154.97 | 262.48 | 450.6 |
| Peak year 2030 rate 0.7 | 61.33 | 89.63 | 145.24 | 236.99 | 407.48 |
| Peak year 2035 rate 0.7 | 70.97 | 100.7 | 163.7 | 289.48 | 481.1 |
| **(b)** | | | | | |
| Peak year 2020 rate 0.3 | 23.65 | 58.85 | 101.21 | 159.63 | 224.06 |
| Peak year 2020 rate 0.5 | 0.1 | 45.02 | 81.11 | 132.74 | 175.4 |
| Peak year 2020 rate 0.7 | −9.61 | 38.47 | 72.75 | 122.58 | 159.76 |
| Peak year 2025 rate 0.3 | 41.76 | 72.05 | 121.28 | 195.42 | 290.79 |
| Peak year 2025 rate 0.5 | 20.46 | 56.46 | 97.95 | 154.76 | 215.79 |
| Peak year 2025 rate 0.7 | 8.57 | 49.88 | 87.93 | 141.32 | 190.25 |
| Peak year 2030 rate 0.5 | 37.16 | 68.74 | 116.01 | 185.94 | 273.78 |
| Peak year 2030 rate 0.7 | 26.57 | 61.05 | 104.71 | 165.36 | 233.41 |
| Peak year 2035 rate 0.7 | 42.68 | 72.78 | 122.86 | 198.09 | 295.71 |

Units of sea-level rise are cm. Units for emission reduction rates are GtC yr$^{-2}$.

rise. Higher emissions in 2030 imply higher long-term sea-level rise (see Fig. 4a). A clear relation between 2050 emission levels (Fig. 4b) and long-term sea-level rise holds for net-zero GHG scenarios. The net-zero CO$_2$ scenarios that have reached zero emissions in 2050 show a difference of 30 cm for median sea-level rise in 2300 (Fig. 4b), which is solely caused by difference in emissions before 2050. This difference reaches >1.5 m for the 95th percentile (Fig. 4e), underscoring the importance of early emission reductions to limit the risk of extreme sea-level rise. Furthermore, we find that a delay of global peak emissions by 5 years in scenarios compatible with the Paris Agreement results in around 20 cm of additional median sea-level rise in 2300 (Fig. 4c). Based on the 95th percentile, we estimate that each 5 years of delay bear the risk of an additional 1 m of sea-level rise by 2300 (Fig. 4f).

**Sea-level legacy of temperature overshoot**. Finally, we assess the sea-level legacy of temporarily exceeding a 1.5 °C warming level[3,23], which is at times referred to as temperature overshoot. The time of overshoot relates quasi-linearly to median sea-level rise in 2300 for net-zero GHG scenarios, adding ~4 cm of median sea-level rise per 10 years overshoot above 1.5 °C (Fig. 5a). Net-zero CO$_2$ scenarios show that without balancing other GHG emissions through negative CO$_2$ emissions, median sea-level rise above 1.5 m is possible for temperature stabilization below 2 °C. No scenario in our set shows median sea-level rise below 1.2 m by 2300 once its global-mean temperatures pass the 1.5 °C level (Fig. 5a, half-filled markers). Another way to look at overshoots is by considering the integrated overshoot temperature over time. When this overshoot integral is larger than 60 °C·yr (Fig. 5b), the difference in year 2300 median sea-level rise between net-zero GHG and net-zero CO$_2$ scenarios disappears. This equivalence can be understood by looking at the characteristic temperature evolution in both scenario subsets (Fig. 1). Net-zero CO$_2$ scenarios consistent with the Paris Agreement exhibit small but persistent overshoot, while net-zero GHG scenarios show higher

but shorter overshoots. At an integral overshoot value larger than 60 °C·yr, the additional early sea-level rise pulse during the high overshoot in net-zero GHG scenarios cannot be compensated through lower rates of sea-level rise later due to declining temperatures until 2300. Only a limited amount of sea-level rise can be avoided through falling temperatures over multiple centuries.

**Sea-level rise and Nationally Determined Contributions (NDCs)**. Our approach allows to link long-term sea-level rise to 2030 emission levels, the current maximum time frame for the NDCs under the Paris Agreement. The NDC emissions aggregated for all countries are estimated to result in a range from 49 to 58 Gt CO$_2$eq yr$^{-1}$ (ref. [24]). This is at the upper range of 2030 emission levels assessed here (full range: 27.5–57.5 Gt CO$_2$eq yr$^{-1}$, Supplementary Data 1). Median sea-level rise in 2300 for the four net-zero CO$_2$ scenarios consistent with the NDCs is between 145 and 164 cm. The four net-zero GHG scenarios consistent with the NDCs show a median sea-level rise in 2300 between 105 and 123 cm. Our 95th percentile estimates indicate a risk of sea-level rise between 4.1 m and 4.8 m for the four NDC-consistent net-zero CO$_2$ scenarios and between 2.3 and 3 m for the four NDC-consistent net-zero GHG scenarios (Supplementary Data 2). These NDC sea-level estimates rely on stylized emission pathways that successfully reach either net-zero CO$_2$ or net-zero GHG emissions in the 21st century based on very ambitious annual reduction rates after 2030. Integrated assessment models (IAMs) suggest that 2030 emission levels implied by current NDCs need to be brought down substantially to achieve the Paris Agreement mitigation and long-term temperature goal[25].

**Discussion**

In this work, we link stylized emission scenarios that reflect the Paris Agreement goals to sea-level rise in the year 2300. Our results indicate that sea-level rise will continue until and beyond 2300 even for scenarios that reach net-zero GHG emissions in the second half of the 21st century. The long-term sea-level legacy under the Paris Agreement scenarios are strongly influenced by emission reductions in the next couple of decades because offsetting sea-level rise in the 22nd and 23rd centuries is hindered by its high inertia, low reversibility and the limited effect of net-zero GHG emissions. By combining climate and sea-level uncertainties, our analysis reveals a persistent risk of high sea-level rise even under pathways in line with the Paris Agreement. However, extreme sea-level rise projections at the 95th percentile of our distribution can be halved through early and stringent emission reductions.

In our scenario set, we vary fossil-fuel and industry CO$_2$ emissions to create scenarios with different characteristics. Climate forcers other than CO$_2$ are co-emitted when fossil fuel is burned[26]. We do not vary co-emissions in our scenarios as the difference through changes in co-emissions is limited given the stringency of each of our scenarios[26]. For the achievement of net-zero GHG emissions levels, residual non-CO$_2$ emissions are balanced by negative CO$_2$ emissions. Variations in these residual non-CO$_2$ GHGs could result in higher or lower rates of temperature decline after peak warming.

A model that makes the sea-level assessment feasible for multiple emission scenarios needs to rely on simplified representations of the underlying processes. This comes with a number of caveats. Our methodology largely builds on the link between sea-level response and global-mean temperature increase. It cannot be ruled out that such link is weak for some components or will get weaker in the future. We have accounted for the potential of self-sustained ice loss that becomes independent of

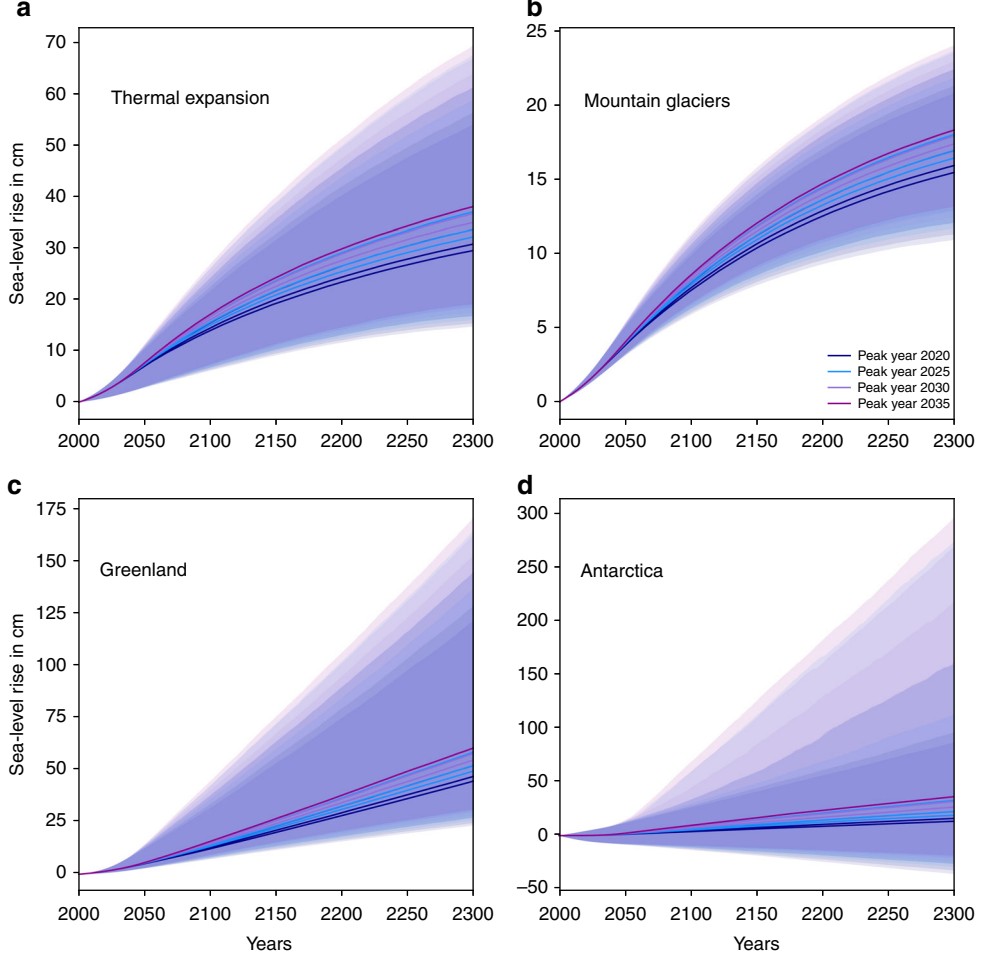

**Fig. 2** Response of the sea-level contributors to net-zero $CO_2$ scenarios. Time series of the sea-level responses of thermal expansion **a**, mountain glaciers **b**, Greenland mass balance **c**, and Antarctic mass balance **d**. Sea-level rise is in cm and relative to the year 2000. Colors refer to peak years as in Fig. 1. Shadings show the central 90th percentile range

global-mean temperature change in our parametrization for the Antarctic ice sheet (Methods section).

For the Greenland ice sheet, potential self-sustained dynamics are not included in our parametrization though feedbacks exist[27, 28] and their future role for Greenland ice loss is not fully clear. The melt-elevation feedback—lower elevation leads to increased melt, which lowers elevation—can lead to threshold behavior and long-term decay of the ice sheet, but it would evolve slowly under the levels of warming assessed here[27]. The melt-albedo feedback[28]—melting exposes darker ice, which absorbs more heat and increases melt—is not yet incorporated fully in process-based simulations. It is also not explicitly modeled in our approach and enters only indirectly through the calibration to recent observations of high-surface melting. Anomalous atmospheric conditions are proposed as the main cause for this recent high melting[29, 30]. This makes it less probable that future ice loss is dominated by the melt-albedo feedback. Our Greenland solid ice discharge parametrization does not account for ice flow instability. The potential for large-scale self-accelerating marine-ice-sheet-instability-driven ice loss like in Antarctica is limited in Greenland as it does not have large marine ice basins that are open to the ocean[31]. Warmer ocean temperatures will still affect ice loss, which is covered in our approach. Our estimate of Greenland's solid ice discharge is above the range of ice-sheet model simulations, but these simulations do not fully resolve outlet glaciers dynamics and do not fully reflect the observed continuing solid ice discharge[20]. Nevertheless, the absence of

feedbacks in the Greenland mass balance representation of our approach is a caveat and we cannot rule out that such feedbacks add to the presented numbers here. Changes in land water storage[32, 33] are not covered in our method as they are not directly linked to climate change. More details are given in Methods section.

Some earth-system response features associated with falling global temperatures in net-zero GHG scenarios may not be captured correctly by the reduced-complexity model MAGICC. In particular, the hemispheric upwelling-diffusion ocean model may not comprehensively simulate the ocean response to sustained atmospheric cooling[34]. However, sea-level rise from ocean warming is calculated by the sea-level model and not directly by MAGICC in our approach.

Due to the slow response to climate warming, sea level will continue to rise after temperatures have stabilized. Positive rates of sea-level rise will likely persist through 2300 even under net-zero GHG emissions and decreasing global-mean temperatures. Stabilizing global-mean temperature rise or achieving net-zero $CO_2$ emissions can only be seen as a first step to halt global sea-level rise. Early peaking and stringent emissions reductions thereafter are vital and important to reduce the risk of low-probability high-end sea-level rise, yet insufficient to stop global sea-level rise by 2300. Delayed near-term mitigation action in the next decades will leave a substantial legacy for long-term sea-level rise.

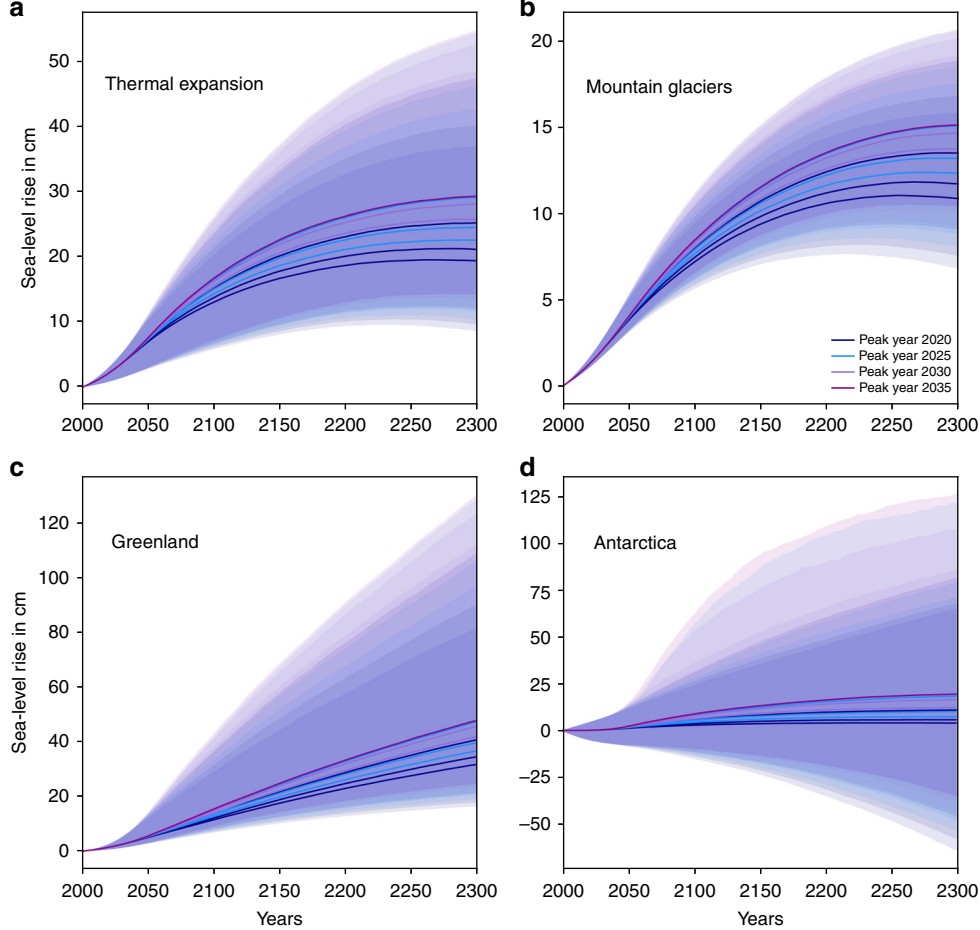

**Fig. 3** Response of the sea-level contributors to net-zero GHG scenarios. Time series of the sea-level responses of thermal expansion **a**, mountain glaciers **b**, Greenland mass balance **c**, and Antarctic mass balance **d**. Sea-level rise is in cm and relative to the year 2000. Colors refer to peak years as in Fig. 1. Shadings show the central 90th percentile range

**Table 2 Rates of sea-level rise at the end of the 22nd century (2290–2300 mean) for (a) zero CO$_2$ and (b) zero GHG scenarios**

| Percentile | 5.0 | 16.66 | 50.0 | 83.33 | 95.0 |
|---|---|---|---|---|---|
| (**a**) | | | | | |
| Peak year 2020 rate 0.3 | 0.12 | 0.2 | 0.39 | 0.74 | 1.15 |
| Peak year 2020 rate 0.5 | 0.11 | 0.17 | 0.36 | 0.59 | 1.05 |
| Peak year 2020 rate 0.7 | 0.07 | 0.16 | 0.33 | 0.58 | 0.88 |
| Peak year 2025 rate 0.3 | 0.15 | 0.26 | 0.46 | 0.88 | 1.6 |
| Peak year 2025 rate 0.5 | 0.11 | 0.2 | 0.39 | 0.74 | 1.15 |
| Peak year 2025 rate 0.7 | 0.12 | 0.2 | 0.37 | 0.65 | 1.16 |
| Peak year 2030 rate 0.5 | 0.16 | 0.26 | 0.45 | 0.83 | 1.45 |
| Peak year 2030 rate 0.7 | 0.12 | 0.22 | 0.43 | 0.78 | 1.55 |
| Peak year 2035 rate 0.7 | 0.17 | 0.25 | 0.49 | 0.94 | 1.76 |
| (**b**) | | | | | |
| Peak year 2020 rate 0.3 | −0.17 | 0.02 | 0.13 | 0.3 | 0.38 |
| Peak year 2020 rate 0.5 | −0.27 | -0.04 | 0.06 | 0.24 | 0.33 |
| Peak year 2020 rate 0.7 | −0.23 | -0.03 | 0.06 | 0.22 | 0.29 |
| Peak year 2025 rate 0.3 | −0.06 | 0.07 | 0.2 | 0.39 | 0.54 |
| Peak year 2025 rate 0.5 | −0.17 | 0.0 | 0.12 | 0.27 | 0.43 |
| Peak year 2025 rate 0.7 | −0.25 | 0.0 | 0.08 | 0.28 | 0.41 |
| Peak year 2030 rate 0.5 | −0.12 | 0.05 | 0.17 | 0.35 | 0.47 |
| Peak year 2030 rate 0.7 | −0.15 | 0.02 | 0.15 | 0.29 | 0.37 |
| Peak year 2035 rate 0.7 | −0.07 | 0.06 | 0.19 | 0.41 | 0.58 |

Units of rates of sea-level rise are cm yr$^{-1}$. Units for emission reduction rates are GtC yr$^{-2}$.

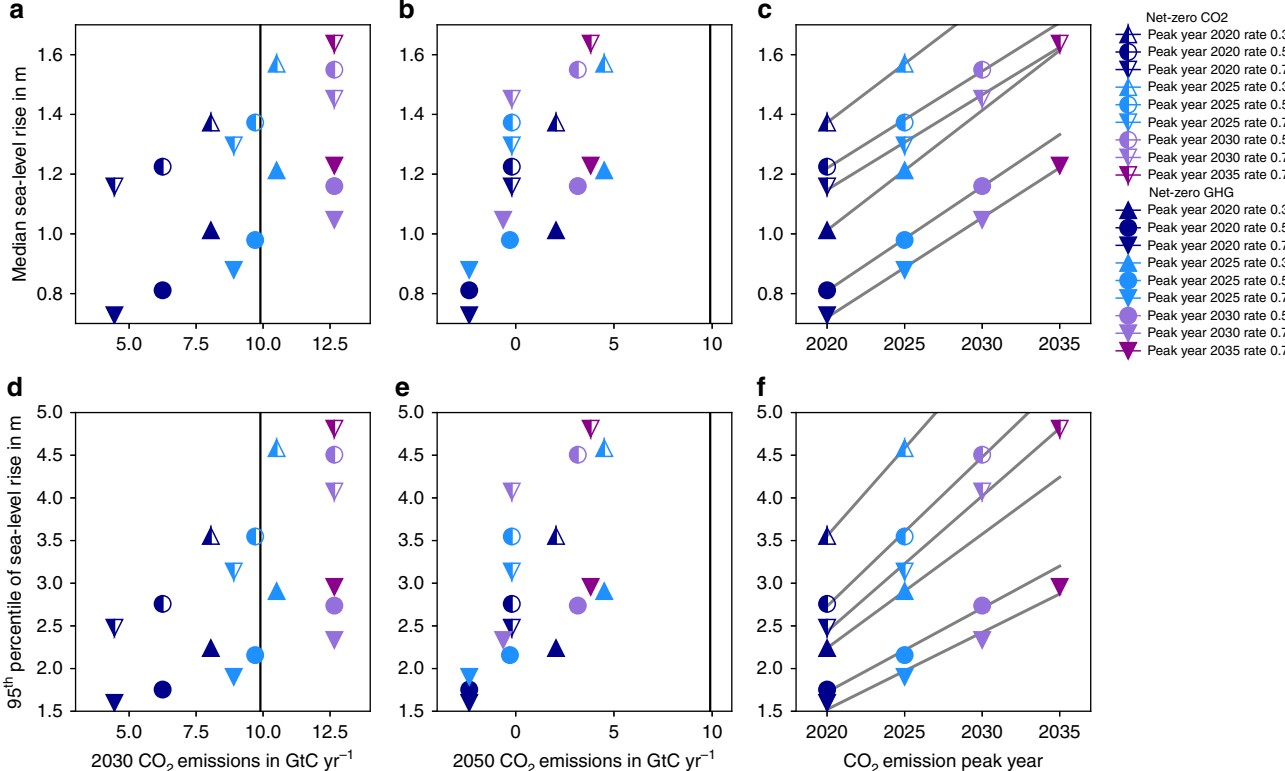

**Fig. 4** Characteristics of emission pathways versus sea-level rise in 2300. $CO_2$ emission levels in 2030 **a**, **d** and 2050 **b**, **e** for net-zero $CO_2$ (half-filled markers) and net-zero GHG (filled markers) scenarios versus median **a**, **b** and 95th percentile **d**, **e** sea-level rise in 2300. The vertical black line shows $CO_2$ emissions in 2015. **c**, **f** $CO_2$ emission peak year versus median **c** and 95th percentile **f** sea-level rise in 2300 with same markers as in **a**–**d** and linear fits for scenarios with the same decarbonization rate as gray lines. Rate in legend refers to the rate of emissions reductions after the emissions peak in GtC yr$^{-2}$

## Methods

**Scenarios**. We construct our scenarios by modifying the harmonized emissions of Representative Concentration Pathway 2.6 (RCP2.6)[11]. This is the only of four RCPs that is close to the requirements set by the Paris Agreement. We update this pathway with observed 2005–2015 fossil-fuel and cement $CO_2$ emissions[35] and then modify fossil-fuel and cement $CO_2$ emissions, while keeping land use $CO_2$ and GHG emissions other than $CO_2$ unchanged. The constructed scenarios, shown in Fig. 1a, d, vary in two parameters: the peak year of fossil-fuel and cement $CO_2$ emissions (henceforth referred to as $CO_2$ emissions) and the reduction rate of those $CO_2$ emissions. We let $CO_2$ emissions increase linearly with the mean 2009–2015 rate until the year of peak emissions (between 2020 and 2040 in steps of 5 years) and linearly decline emissions after the peak until net-zero $CO_2$ (Fig. 1a) or net-zero GHG emissions (Fig. 1d) are reached. Negative $CO_2$ emissions compensate residual land-use $CO_2$ emissions in the net-zero $CO_2$ scenarios and compensate land-use $CO_2$ plus all other residual GHG emissions of the RCP2.6 scenario in the net-zero GHG scenarios. For global-mean temperature and sea-level responses to the original RCP2.6 scenario see Supplementary Fig. 1 and Supplementary Data 3. We here use the 100-year global warming potential (GWP-100) from the Second Assessment Report of the IPCC[24, 36] to estimate the amount of $CO_2$ needed to offset the other emissions of the RCP2.6 scenario.

**Global-mean temperature projections**. We apply the reduced-complexity climate and carbon cycle model MAGICC[16, 17] to determine the climate system response to the net-zero $CO_2$ and the net-zero GHG scenarios (Figs. 1b, e, respectively). To cover the uncertainty, we sample from 600 sets of climate and carbon cycle parameters, which are constrained through past climate change and climate models of higher complexity. Scenarios not in line with the criterion of holding warming below 2 °C with a likelihood of at least 66% are excluded (gray lines in Fig. 1). The net-zero $CO_2$ emissions scenarios show temperature stabilization (Fig. 1b). In these scenarios decaying atmospheric $CO_2$ concentrations are balancing the evolution from the transient to the equilibrium temperature response, while non-$CO_2$ GHGs (dominated by shorter lived GHGs like $CH_4$) are approximately constant and thus result in a constant non-$CO_2$ temperature contribution. Scenarios with net-zero GHGs result in declining temperatures (Fig. 1e), because atmospheric $CO_2$ concentrations decline faster than in the net-zero $CO_2$ case, and all other contributions are kept the same[37].

**Global sea-level projections**. To project future sea-level rise, we apply a component-based global sea-level model[18], which includes an updated

parameterization of the Antarctic ice sheet response based on refs. [19, 38,]. For thermal expansion, glaciers and the Greenland surface mass balance, the model combines the long-term sensitivity[15] of each component to global-mean temperature warming with the individual recent observations. Future evolution is thus constrained to both long-term sensitivity and past observations. For the three components, we use a pursuit curve approach, in which the difference between long-term sensitivity $S_{eq}(T, \alpha)$ and the time-dependent contribution $S(t)$ drives the rate of sea-level rise for each component:

$$\frac{dS}{dt} = \frac{S_{eq}(T(t), \alpha) - S(t)}{\tau}. \quad (1)$$

The sensitivity parameter $\alpha$ is determined from equilibrium simulations for each component, with uncertainty in the long-term sensitivity covered by variation of the parameter. $S_{eq}(T, \alpha)$ has different functional forms for the different components. The response time $\tau$ is calibrated to observations for each component, with the range of $\tau$ reflecting the uncertainty in observations. We use a response-function approach[39, 40] for the Greenland solid ice discharge due to missing long-term sensitivity estimates or past trends in observations. This assumes that frontal stress release[41] and runoff lubrication[42] can be approximated as linearly depending on the global-mean temperature anomaly (ref. [18], equation 4). Both Greenland surface mass balance and solid ice loss are calibrated to new observations[20, 21]. To cover the recently proposed increased sensitivity of the Antarctic ice sheet to global warming[19], we update our method to capture the corresponding Antarctic sea-level contribution. We utilize a parametrization that combines a slow and gradual response to global warming with a fast discharge term that mimics ice instability[38]. The parametrization has four free parameters that include a threshold temperature and a rate for fast discharge. Fast discharge contributes to sea-level rise once the threshold temperature is passed. We create a 29-member ensemble of the four-parameter set by calibrating our parametrization to the response of each ensemble member available from ref. [19] for the RCP2.6, RCP4.5, and RCP8.5 scenarios.

The long response times of ice sheets, glaciers and ocean make it probable that a fraction of their contribution originates from their ongoing adaptation to past climate change, i.e., the little or the last ice age[5, 6, 10, 43]. Only for glaciers we are able to explicitly incorporate such a natural fraction, which is declining and below 30% at present[43]. We do not include changes in land water storage as climate mainly influences its decadal variability[44] and not its long-term trend. Direct human influence on land water storage through groundwater pumping[33, 45] and

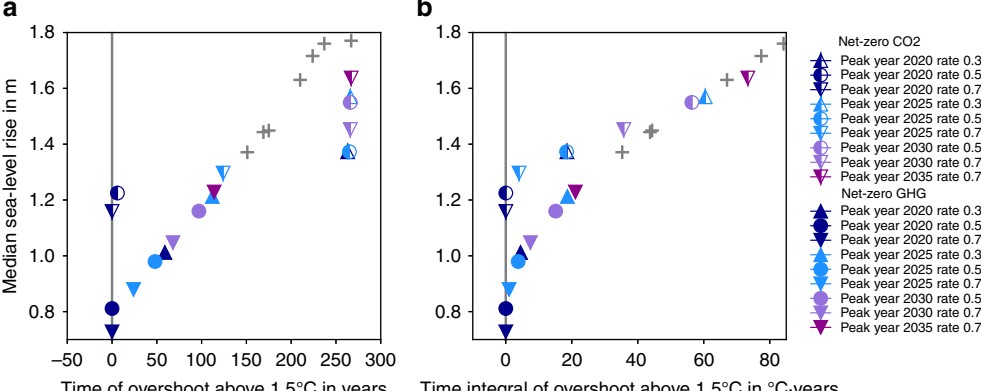

**Fig. 5** Sea-level rise in 2300 and temperature overshoot above 1.5 °C. **a** Median sea-level rise versus the temporal overshoot in years. **b** Median sea-level rise versus the time integral of the temperature overshoot. Half-filled markers indicate net-zero $CO_2$ scenarios, filled markers net-zero GHG scenarios. Gray crosses show net-zero GHG scenarios that do not comply with the 2 °C target of the Paris Agreement. Rate in legend refers to the rate of emissions reductions after the emissions peak in GtC $yr^{-2}$. The grouping of net-zero $CO_2$ scenarios exceeding 1.5 °C in the upper-right corner of **a** reflects the length of our simulations and is of limited significance

dam building[32] cannot be linked to global climate change and would thus unnecessarily blur our analysis.

Our aggregate uncertainty estimates are based on Monte-Carlo sampling: for each sea-level contribution, we draw from the calibrated sets of sea-level functions, which incorporate the different observational datasets and long-term estimates, and from the 29 calibrated parameter sets of the Antarctic component. We drive the selected sea-level functions with a global-mean temperature pathway, randomly drawn from the 600 member global-mean temperature ensemble for a specific scenario. The sampling is repeated 10000 times.

**Individual sea-level components and comparison to literature.** Thermal expansion: We estimate thermal expansion with the constrained-extrapolation approach following eq. 1, driven by the global-mean temperature evolution estimated by MAGICC[16, 17]. The comparison of our thermal expansion estimate for the year 2300 for the RCP2.6 scenario (median: 26 cm; central 66th percentile range: 17–36 cm, relative to 2000, Supplementary Data 3) can inform on the performance of our model. Other studies have reported 2300 steric sea-level rise of 6–37 cm relative to 1986–2005 for an ensemble of models of intermediate complexity[46] and 14–27 cm relative to 2006 for a set of six CMIP5 models[47]. Sea-level rise through thermal expansion is thus for RCP2.6 comparable to more complex models. In line with CMIP5 model experiments[48], our steric sea-level rise component does not exhibit a decline in sea-level under the net-zero GHG scenarios until 2300, despite the projected global-mean temperature decline. Median sea-level rise ranges from 30 to 38 cm in 2300 for net-zero $CO_2$ scenarios and from 19 to 30 cm for net-zero GHG scenarios (Supplementary Data 4).

Mountain glaciers: We estimate the contribution from mountain glaciers following equation 1. Our estimate for the median contribution from glaciers until year 2300 is 15.9 cm for the RCP2.6 scenario (Supplementary Data 3). Median sea-level rise from glaciers ranges from 15.5 to 18.3 cm in 2300 for our net-zero $CO_2$ scenarios and 11 to 15.2 cm for net-zero GHG scenarios (Supplementary Data 4).

Greenland ice sheet: We estimate the contribution from Greenland's surface mass balance following eq. 1. We use a response-function approach[39, 40] for the Greenland solid ice discharge due to missing long-term estimates or past trends in observations. This assumes that frontal stress release[41] and runoff lubrication[42] can be approximated as linearly depending on the global-mean temperature anomaly (equation 4 in ref. [18]). We recalibrate the Greenland surface mass balance and solid ice discharge parameterizations to updated observations[20, 21]. These observations include the recent years of strong mass balance changes and lead to a larger spread in our calibrated parameters. Calibrated parameters are listed in Supplementary Data 5. The spread is reflected in higher 95th percentile estimates for both surface mass balance and solid ice discharge as compared to ref. [18]. Median sea-level rise from the combined Greenland surface mass balance and solid ice discharge ranges from 45 to 61 cm in 2300 for our net-zero $CO_2$ scenarios and 32–48 cm for net-zero GHG scenarios (Supplementary Data 4).

Greenland has only limited marine-grounded regions that are open toward the ocean, ice drains predominantly through narrow fjords. The potential for large-scale self-accelerating marine ice sheet instability hence is also limited, in contrast to the Antarctic ice sheet. Warmer ocean temperatures will still affect ice loss, but this contribution is covered in our response-function approach. The approach yields continuing Greenland solid ice discharge throughout 2300 (median for RCP2.6 scenario: 23.5 cm, Supplementary Data 3). This differs from earlier observations of reduced discharge[49] and ice-sheet simulations showing a diminishing contribution from solid ice discharge[50–52]. These process-based simulations can however not fully resolve the narrow outlet glacier flow to the

ocean that is central for solid ice discharge. They do not incorporate the deeper and larger subglacial basins[31] and deeper fjords[53] in new ice and ocean floor data, which suggest increased sensitivity to climate warming. The broad range (16–92 cm in 2300 for the RCP2.6 scenario, Supplementary Data 3) reflects at least partly our incomplete knowledge of this term.

Two feedbacks related to the surface mass balance may foster a self-sustained decay of the Greenland ice sheet. The melt-elevation feedback[27] can add to the directly temperature-driven sea-level contribution of our approach. In the model of ref. [27] the decay of the ice sheet occurs on a time scale longer than 10,000 years for 2 °C temperature increase over Greenland (their Fig. 3). In the SRES A1B scenario[54], a scenario without any climate policy and estimated year-2100 warming of 3–4 °C[55], the ice loss through this feedback has been estimated to be between 3.6 and 16% of the forcing-driven ice loss in 2200 (ref. [56]). Relating this fraction to our scenarios with lower warming is however not straightforward. Still, both ref. [27] and ref. [56] show the bounded role of the feedback. We thus do not see the melt-elevation feedback sufficiently large to dismiss or invalidate our approach on the timescales and warming levels assessed here. In three recent publications process-based ice sheet models have been used to project future Greenland ice loss. Ref. [50] find a total (surface-mass-balance and ice-dynamic) contribution of 9 cm under RCP2.6 until 2300. Ref. [52] find 10 cm surface mass balance contribution from Greenland until 2100 under the RCP4.5 scenario. Ref. [57] find 4 (2–6) cm for the same scenario until 2100. The melt-albedo feedback[28] is not fully integrated in these studies and may render these model results too low. This feedback has a physical (less snow and more bare ice absorb more radiation) and a biological component (wetter conditions allow more ice surface algae growth[30, 58]). There are currently no process-based simulations available that estimate future Greenland ice loss while fully including this feedback. Ref. [28] shows that the extreme 2012 conditions had similar low ice albedo conditions as projected for the end of the century, and therewith highlighted that models are still incomplete for such projections. Assuming that yearly repeated 2012 conditions can be used as an upper bound for climate scenarios that stay below 2 °C, the upper bound of 670 Gt mass loss in 2012 would lead to 56 cm sea-level rise when summed up for 300 years. This is about 13 cm above our 95th percentile surface mass balance estimate for Greenland for RCP2.6. If the melt-albedo feedback becomes a major driver of Greenland ice loss, these values could be exceeded. Research however now suggests that while the melt-albedo feedback enhances the ice loss, changes in the atmospheric circulation are the ultimate driver of the high melting in recent years[29, 30]. A runaway feedback between atmospheric changes and the Greenland melt is not evident, which makes a self-sustained ice sheet collapse (as compared to climate-forcing driven collapse) less probable. Median sea-level rise from the combined Greenland surface mass balance and solid ice discharge ranges from 45 to 61 cm in 2300 for our net-zero $CO_2$ scenarios and 32–48 cm for net-zero GHG scenarios (Supplementary Data 4).

Antarctic ice sheet: We apply a parametrization for Antarctic mass loss[38], which incorporates increased sensitivity to global warming through two newly proposed instability mechanisms[59]. The mechanisms suggest a tight link between future atmospheric warming and Antarctic ice discharge[19]. The discharge thus also depends on the emission scenario. Our parametrization is calibrated to the results of ref. [19] and combines a slow and gradual response to global warming with a fast discharge term that mimics ice instability. Once a trigger temperature of 1.9–3.2 °C global-mean temperature rise is reached, the fast discharge adds to sea-level rise at a constant rate of 2–20 mm per year. Technical details are available at https://github.com/matthiasmengel/fast_ant_sid. Median sea-level rise from the Antarctic ice sheet ranges from 13 to 36 cm in 2300 for our net-zero $CO_2$ scenarios and 4–19 cm for net-zero GHG scenarios (Supplementary Data 4).

Ice sheet loss through the marine ice-sheet instability, which is initiated by warmer ocean waters, may not be fully covered by our method. Such instability may already be underway in West Antarctica[60–62]. The instability is difficult to directly link to anthropogenic climate change. While ref. [62] does not provide rates of sea level rise for the main phase of the collapse, simulations for West Antarctica as a whole indicate an upper bound of 5 cm in the first 200 years[63]. The risk of ocean-driven marine-ice-sheet instability hence increases uncertainty in future sea level rise, but the numbers available from process-based simulations suggest a minor role for the timescale considered here. Once triggered and independent of the forcing, it would not affect relative changes between scenarios.

**Code availability**. The sea-level code is available at https://github.com/matthiasmengel/sealevel with the version used in the presented analysis archived at https://doi.org/10.5281/zenodo.1118288. The MAGICC model is not open-source, but a compiled version can be obtained from the authors.

**Data availability**. All data to reproduce the presented analysis are available from https://doi.org/10.5281/zenodo.1116918.

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

## Acknowledgements

M.M. is supported by the AXA Research Fund Postdoctoral Fellowship Programme. J.R. received funding from the European Union's Horizon 2020 research and innovation programme under grant agreement No. 642147 (CD-LINKS) and grant agreement No. 641816 (CRESCENDO) and acknowledges support by the Oxford Martin School Visiting Fellowship Programme. C.-F.S. acknowledges support by the German Federal Ministry for the Environment, Nature Conservation and Nuclear Safety (16_II_148_Global_A_IMPACT).

## Author contributions

M.M and C.-F.S. designed the research. M.M. carried out the modeling and coordinated the research. J.R. led the scenario design. All authors contributed to the discussion and interpretation of the results and to the writing of the paper.

## Additional information

**Competing interests:** The authors declare no competing financial interest.

