## [Peer Review File · Nature Communications]

Reviewers' comments:

Reviewer #1 (Remarks to the Author):

Mengel et al, The sea level legacy of the Paris Agreement and the effect of delayed mitigation action, applies existing models and methods to address a problem of considerable current relevance to the policy world: What do the known lags in the response of sea level to emission and temperature targets proposed under the Paris agreement imply for long term (up to year 2300) projections of sea level rise? Only relatively low emissions scenarios consistent with Paris constraints are considered.

My overall judgment is that this submission needs a significant overhaul to make it useful and once this has been accomplished, probably should be submitted elsewhere.

This paper adds to a body of literature on the consequences of implementing Paris, or failing to do so. None of the previous analyses combine semi-empirical and process based modeling in quite the way the current paper does in this particular application. The question is whether it adds enough to merit publication at this time. For the reasons given below, I would rather see the authors take additional time to expand the analysis and submit a longer version to Nature Climate Change. As written, the paper is too limited by its assumptions, especially with regard to the long-term ice sheet contribution to sea level rise, to make a significant contribution. Furthermore, the English usage is surprisingly odd in a few places (given the authors' competence). The figures also are incomplete in a few respects. A thorough re-read is in order.

The main contribution of the paper is quantification of sea level rise over 300 years in the Paris context. But projections beyond 2100 are only as good as the representation of Antarctic ice sheet dynamics. As the authors (finally) point out near the end, the model used cannot possibly capture ice sheet instabilities that dominate other estimates of future behavior, such as those of DeConto and Pollard (2016) and Joughin et al 2014. The dynamical component becomes progressively more important as the projection goes further into the future, so that the value of the paper is severely limited by its modeling approach. The differences between scenarios pale in comparison to this uncertainty.

The authors could have declared this limitation at the beginning and their reluctance to do so is quite puzzling. Many others have pointed out that the ice dynamic term is the major uncertainty toward the end of the 21st century and beyond. If they had done so, the deficiency of the approach would at least be glaringly obvious. But doing so would not cure the problem. Only an attempt to incorporate such post-AR5 ice sheet estimates into the modeling approach would remedy the problem and make the paper useful. Likewise, the paper is misleading because its uncertainty analysis makes no attempt to quantify this aspect of uncertainty. There are many ways to do so, ranging from sensitivity tests to probabilistic sampling of ice sheet projections. In the sea level context, all of these have been applied by others and these authors are fully capable of doing so. This will lengthen the paper and make it more useful, which is why I propose NCC as the next stop for a different version.

Specific Comments:

Abstract: the two sentences beginning "We estimate median sea level rise..." would be puzzling to a reader who does not understand that Paris contains several distinct long-term objectives.

Line 34: The first of several language oddities - "subsidy" should be "subsidence". There are others and the authors should look out for them.

Lines 68-70: the reader could easily get confused between sea level rise rate and cumulative sea level rise. Some rewording is in order.

Line 95-96: this is one of many findings for 2300 that are simply not of interest given the limitations of the model noted above. The authors highlight several such differences between trajectories that are of minor importance compared to the large ice dynamics uncertainty.

Line 110-113: the reasoning here is unclear.

Line 133-135: Assuming the authors (or editors) take my advice and go back to a much more complete uncertainty analysis, the difference between MAGICC and GCM projections would be worth estimating as well.

Line 139-141: this section is totally misleading for the reasons above. The authors try to imply that the outcomes for ice dynamics and other contributors are somehow well anchored before admitting later on that at least the ice dynamic term is not. This compounds the failure to mention the problem at the outset.

Line 164-66: here we finally have a statement that encapsulates the problem.

Figures: what does "peaky" mean? Peak year? Where does upward dark blue triangle appear on the figures? Why no legend explaining solid line colors in Figure 2?

Reviewer #2 (Remarks to the Author):

Review of: The sea level legacy of the Paris Agreement and the effect of delayed mitigation action

Submitted to Nature Communications by Mengel et al.

Overall this is a solid contribution, but I don't believe it elevates to a level worthy of publication in Nature Communications. The paper certainly does represent an incremental contribution to an important and hot topic—emissions associated with Paris temperatures pledges, and their implications for long term sea level rise given all the research community has been learning about irreversibility and inertia of the ice sheets especially. Unfortunately the paper only mentions all the recent advances in our understanding of ice sheets and irreversibility; the actually modeling and sea level rise projections rely on techniques that are not particularly fresh. The results similarly are straightforward extensions of what is already known.

In terms of the paper's advances, albeit incremental, to my knowledge the discussion of how a delay in emissions peak year impacts 2300 sea level rise is previously unreported (in the context of Paris targets), as is the discussion of how much CO₂ overshoot (in the context of Paris targets) impacts 2300 sea level rise. However, the answer to the former is pretty unsurprising (a 15 year delay might mean approximately an additional foot of total sea level rise by 2300) and the latter is barely discussed in the body of the manuscript. More fundamentally, it is becoming clearer that there are real limitations in the modeling frameworks the authors use, such as MAGICC and the IPCC AR5 sea level rise projections. While the authors do an adequate job of mentioning that these limitations exist, from my perspective a Nature Communications paper would need to a) actually incorporate into their analysis the latest understanding of how ice sheets may change (e.g Deconto and Pollard 2016, Bamber and Aspinall 2013 and a bevy of other post AR5-cutoff date papers) or b) weigh in on why their approach—based on a simplified set of assumptions--remains justified despite all the emerging science.

Again, I believe the paper with some minor work (see below) will deserve publication in a specialist journal, where it is common to make a set of simple assumptions based on models that have an established history such as MAGICC, and apply those models with a new angle (e.g. Paris targets, and 2300 projections). I think a Nature Communications paper would need to more deeply sample the true range of possible outcomes using this new angle (Paris targets, and 2300 projections). The absence of any real surprises in the results further undermines the case for publication in Nature Communications.

Below are minor comments that might help the authors should they seek publication in a specialist journal, where I am confident this paper will find a good home.

Line 34: 'subsidy' to 'subsidence' ?

Line 35: Use a crisper reference here, such as Sweet and Park, 2014.

Line 43: Change citation to Jevrejeva et al.

Lines 47-49: this is not a sentence

Line 65-66: You have not yet said what the baseline sea level period is.

Lines 78-79 and elsewhere: consider changing 'until year 2300' to 'through at least the year 2300' just to ensure no one thinks you are saying it changes in 2300

Line 115: There are grammar issues in the second half of this sentence.

Line 171: should read 'have yet'

Other comments

The differences between, and assumptions being made about, net zero GHG vs. net zero CO₂ that is obliquely explained in lines 103-105 should be more directly explained at first use. It is hard to follow in the current version of the manuscript.

It strikes me as a bit awkward to have Figures 1a and 1b not called out until the 'end' of the paper (methods). Perhaps it is OK, depending on the journal; you might just think about pros and cons, and whether there is a better option.

The paper mentions a sufficient number of caveats, but for at least a couple of the biggies, you might add another sentence saying something about it. For example, it may be really important that the IPCC AR5 projections did not include Marine Ice Sheet Instability (even if we do not exceed 2C warming). As another example, is there an argument against trusting your Greenland reversal findings? At least a little discussion based the literature would seem to be justified here.

References

Bamber, J. L., and W. P. Aspinall (2013), An expert judgement assessment of future sea level rise from the ice sheets, *Nat. Clim. Change*, 3, 424–427, doi:10.1038/nclimate1778.

Sweet WV, Park J. From the extreme to the mean: Acceleration and tipping points of coastal inundation from sea level rise. *Earth's Future*. 2014;2(12):579-600. doi: 10.1002/2014EF000272.

Reviewer #3 (Remarks to the Author):

Overall this is an excellent and well-written paper. It addresses the issue of sea level commitment. The results of the paper will be of wide interest to a range of academics, policy makers and coastal stakeholders. I recommend publication if the following moderate-issues below are addressed:

1. I think the final paragraph introducing the methods on line 51-58 needs to be expanded. Currently it is too brief. I think you need to add a sentence or two outlining what scenarios you have run. On Line 67 you say 'for the three scenarios' but I am not sure until the methods section what these scenarios are! Please include a briefly description in the methods paragraph above regarding this.
2. This is just a suggestion, but I recommend showing a sub-plot on Figure 1, beneath panels c and f, which show the rate of sea level rise per year. This will make it easier to see when things start flattening off.

3. I think it would be good to stress in either the introduction or conclusion section the importance of considering the commitment of sea level rise in coastal planning. For example major schemes, such as the Thames Barrier, or the building of Nuclear power stations, with a life-time of more than 100 years, must account for sea level commitments beyond 2100.

4. In the discussion the results of this study are not compared/contrasted with other efforts that have projected sea level out beyond 2100. I think a paragraph should be added describing briefly how the authors results agree with other published efforts, particularly from GCMs, to project sea level rise out beyond 2100.

5. The method section in my opinion is way to brief. Although the model used to project future sea level rise is described in another paper (Mengel et al., 2016) I still think it is important here in this current paper, that it is at least described in some way. Could I encourage the authors to add another paragraph briefly describing how the contribution-based semi-empirical model works.

AUTHOR RESPONSES TO THE REFEREE COMMENTS

Manuscript ID: NCOMMS-17-00342

Title:

The sea level legacy of the Paris Agreement and the effect of delayed mitigation action

Authors:

Matthias Mengel, Alexander Nauels, Joeri Rogelj, Carl-Friedrich Schleussner

Dear Referees,

Hereby we send you the revised version of our manuscript with number “NCOMMS-17-00342”. We would like to thank the three referees for their constructive remarks and very helpful suggestions. In particular, the reviewers highlighted important issues related to recent scientific advances in our understanding of the potential contribution of Antarctic ice instabilities. We fully agree that a substantive discussion of this issue was lacking in our initial submission and that an analysis of such instabilities is vital for the credibility of our assessment. We have responded to this critique in full, as well as to all other issues raised. Thanks to these revisions, we are now pleased to submit a significantly revised and improved manuscript that is much clearer positioned and shows the robustness of our findings in relation to the recent literature in this highly dynamic field.

Point-by-point responses to the referee comments are inserted below in blue. Quotes from the revised manuscript are in red.

Yours sincerely,

Matthias Mengel

For the author team

Reviewer #1 (Remarks to the Author):

Mengel et al, The sea level legacy of the Paris Agreement and the effect of delayed mitigation action, applies existing models and methods to address a problem of considerable current relevance to the policy world: What do the known lags in the response of sea level to emission and temperature targets proposed under the Paris agreement imply for long term (up to year 2300) projections of sea level rise? Only relatively low emissions scenarios consistent with Paris constraints are considered.

My overall judgment is that this submission needs a significant overhaul to make it useful and once this has been accomplished, probably should be submitted elsewhere.

RESPONSE: Thank you very much for this critical review. We agree that our initial submission left out important new literature concerning Antarctic ice sheet dynamics. This was a clear gap in our article and a disservice to the interested reader trying to achieve an integrated picture of sea level rise under Paris Agreement scenarios. We have therefore fully taken into account these constructive suggestions in the presented revisions (see responses below).

This paper adds to a body of literature on the consequences of implementing Paris, or failing to do so. None of the previous analyses combine semi-empirical and process based modeling in quite the way the current paper does in this particular application. The question is whether it adds enough to merit publication at this time. For the reasons given below, I would rather see the authors take additional time to expand the analysis and submit a longer version to Nature Climate Change. As written, the paper is too limited by its assumptions, especially with regard to the long-term ice sheet contribution to sea level rise, to make a significant contribution. Furthermore, the English usage is surprisingly odd in a few places (given the authors' competence). The figures also are incomplete in a few respects. A thorough re-read is in order.

RESPONSE: Thank you. We clearly agree with the need to provide a comparison of our estimates with potential long-term ice sheet contributions and have done so now in our revised manuscript (see further below). In response to the remarks by the referees, large parts of the manuscript have been rewritten or substantially edited. We apologize for the mistakes in language and carefully attempted to avoid such mistakes in the revised version.

The main contribution of the paper is quantification of sea level rise over 300 years in the Paris context. But projections beyond 2100 are only as good as the representation of Antarctic ice sheet dynamics. As the authors (finally) point out near the end, the model used cannot possibly capture ice sheet instabilities that dominate other estimates of future behavior, such as those of DeConto and Pollard (2016) and Joughin et al 2014. The dynamical component becomes progressively more important as the projection goes further into the future, so that the value of the paper is severely limited by its modeling approach. The differences between scenarios pale in comparison to this uncertainty.

The authors could have declared this limitation at the beginning and their reluctance to do so is quite puzzling. Many others have pointed out that the ice dynamic term is the major uncertainty toward the end of the 21st century and beyond. If they had done so, the deficiency of the approach would at least be glaringly obvious. But doing so would not cure the problem. Only an attempt to incorporate such post-AR5 ice sheet estimates into the modeling approach would remedy the problem and make the paper useful. Likewise, the paper is misleading because its uncertainty analysis makes no attempt to quantify this aspect of uncertainty. There are many ways to do so, ranging from sensitivity tests to

probabilistic sampling of ice sheet projections. In the sea level context, all of these have been applied by others and these authors are fully capable of doing so.

RESPONSE: Understanding and quantifying the dynamical component of the Antarctic ice sheet has risen to prominence in the past five years. Several studies show significant potential contributions of the Antarctic ice sheet over the 21st century and beyond. It is important, however, to compare like with like and to understand which sea level rise projections are implied by which forcing. Mixing and matching internally inconsistent results is undesirable, and we agree with the reviewer that quite some confusion might already exist related to these findings that have emerged in the recent literature. We therefore thank the reviewer for pointing out this gap in our article, and have now included a detailed comparison and discussion of this issue.

In particular, we start by highlighting the issue in the introduction of our manuscript and follow this up with a dedicated and detailed discussion in the supplementary text. A summary of the key points is given in the main-text discussion. We made this split to keep the manuscript concise, but are willing to merge the supplementary text into the main text if the reviewer finds this necessary. We provide a summary in the discussion section:

“Our method does not cover processes that become independent of the climate forcing, such as ice instabilities. Marine ice-sheet instability may be already under way (Rignot et al. 2014; Favier et al. 2014; Joughin, Smith, and Medley 2014), but its impact on sea level rise until year 2300 remains limited (Feldmann and Levermann 2015). Newly-proposed instabilities triggered by atmospheric warming (Pollard, DeConto, and Alley 2015) strongly increase Antarctica’s contribution for medium-to strong warming scenarios (DeConto and Pollard 2016). For the strong-mitigation scenarios similar to RCP2.6 as assessed here these instabilities are unlikely to be triggered and our estimate of RCP2.6 is similar to the numbers of (DeConto and Pollard 2016). Importantly, our method can not be expanded beyond the 23rd century as sea level uncertainty through Antarctic ice instability will become more pronounced. [...] See the Supplementary Information for an in-depth discussion of these issues including comparison to results from more complex models.”

This will lengthen the paper and make it more useful, which is why I propose NCC as the next step for a different version.

RESPONSE: Thank you for your valuable suggestions. We agree that our methodology would need to be amended if it were to be used for high emission scenarios and/or time horizons beyond 2300. We are however confident that it is well-suited for the here presented analysis, i.e. for strong mitigation scenarios and a time horizon no longer than 2300. We therefore can present a revised version within the formatting guidelines of *Nature Communications*. Thanks to the points raised by the reviewer, we are now much clearer on the reasons for applicability of our model and approach.

Specific Comments:

Abstract: the two sentences beginning “We estimate median sea level rise...” would be puzzling to a reader who does not understand that Paris contains several distinct long-term objectives.

RESPONSE: Thank you. We have edited the abstract so that it is easier to understand. The sentences now read:

“ We estimate median sea level rise between 68 and 102 cm if net zero greenhouse gas emissions are maintained until 2300, which in our scenarios requires sustained net negative CO₂ emissions. In the absence of such net negative CO₂ emissions, only temperature stabilization at 1.5°C or below keeps sea level rise until 2300 under one meter in our model. “

Line 34: The first of several language oddities - "subsidy" should be "subsidence". There are others and the authors should look out for them.

RESPONSE: We apologize for this typo. We have revised the entire manuscript, including the Supplementary Information to avoid them as much as possible in this revised version.

Lines 68-70: the reader could easily get confused between sea level rise rate and cumulative sea level rise. Some rewording is in order.

RESPONSE: We have connected both statements so that it should be clear that they refer to different aspects of sea level rise. We now write:

"Although sea level rises continuously in all emission pathways through at least year 2300, this is not the case for the rate at which sea level changes. The rate of sea level rise starts to slow down shortly after emissions peak and continues to decline thereafter (Fig. S2). "

We added the supplementary figure (Fig. S2) that shows the rates of sea level rise.

Line 95-96: this is one of many findings for 2300 that are simply not of interest given the limitations of the model noted above. The authors highlight several such differences between trajectories that are of minor importance compared to the large ice dynamics uncertainty.

RESPONSE: Also here one has to compare like with like. The reviewer's comment is correct for scenarios of medium to strong warming, in which ice sheet decay may be initiated from warmer atmospheric conditions (DeConto and Pollard 2016, DP16). The new findings of DP16 play however a minor role for the scenarios discussed here with strong climate mitigation similar to RCP2.6. As ice instabilities are not triggered, the DP16 Antarctic contribution to sea level rise is low and comparable to our approach. The supplementary text details the point, with a summary in the main-text discussion (see above). The SI paragraph reads:

" [...] two newly-proposed instability mechanisms (Pollard, DeConto, and Alley 2015) suggest a tight link between future atmospheric warming and Antarctic ice discharge (DeConto and Pollard 2016). Such discharge thus depends on the emission scenario, with an estimated sea level contribution for medium and strong warming scenarios (RCP4.5 and RCP8.5) that is much higher than in earlier estimates (Church et al. 2013; Golledge et al. 2015; Mengel et al. 2016). These instability mechanisms, which are linked to atmospheric warming, are improbable to be triggered for the scenarios we assess here with similar or less global warming than RCP2.6. Our median 2300 sea level estimate for RCP2.6 for Antarctica (14.5cm, Table S3) is similar to (DeConto and Pollard 2016) (13.5 or 15.0 cm, depending on their Pliocene sea level target, linearly interpolated between the 2100 and 2500 values provided in the publication). Our method does however not fully incorporate the uncertainty related to these instabilities: our 1 sigma estimate (68th percentile) is 10cm and 20 cm below the (DeConto and Pollard 2016) estimate, depending on their Pliocene target. Importantly, our method can not be expanded beyond the 23rd century as sea level uncertainty through ice instability will become more pronounced. "

As pointed out by the reviewer, ocean-induced West Antarctic ice sheet collapse may already be underway, with an unclear relation to climate change. Such process, becoming independent of the forcing once triggered, is not captured by our approach and we regret if this was not fully clarified in the first version of our manuscript.

Contrary to the assertion of the referee, the potential contribution of such a collapse on the time scales considered in our study is small. Even in simulations with maximum ice loss in (Feldmann and Levermann 2015), the collapse adds 7.5 cm sea level rise from West Antarctica within the next 300

years (personal communication, we reference the published 200-year value of 5cm in the manuscript). This upper estimate is below our range given for the sea level contribution from a delay of peak emissions of 5 years (10cm-14cm, line 95-96 in original submission). We are therefore of the view that omitting this potential contribution does not fundamentally disqualify our results. We now clarify in the discussion:

“Our method does not cover processes that become independent of the climate forcing, such as ice instabilities. Marine ice-sheet instability may be already under way (Rignot et al. 2014; Favier et al. 2014; Joughin, Smith, and Medley 2014), but its impact on sea level rise until year 2300 remains limited (Feldmann and Levermann 2015).”

And we add more detail in the supplementary text:

“West Antarctic ice sheet collapse may already be underway, induced by warmer ocean waters (Mouginot, Rignot, and Scheuchl 2014; Favier et al. 2014; Joughin, Smith, and Medley 2014; Feldmann and Levermann 2015) . Such collapse, which can become independent of its trigger, is difficult to link to present anthropogenic climate change as well as future climate pathways. Even under the strongest applied forcing, a West Antarctic ice sheet collapse is not unfolding for centuries (Joughin, Smith, and Medley 2014). While Joughin, Smith, and Medley (2014) do not provide rates of sea level rise for the main phase of the collapse, whole West Antarctica simulations indicate an upper bound of 5 cm in the first 200 years (Feldmann and Levermann 2015). The risk of ocean-driven marine-ice-sheet instability hence increases uncertainty in future sea level rise, but plays a minor role for the time scale considered here. Once triggered and independent of the forcing, it would not affect relative changes between scenarios.”

Line 110-113: the reasoning here is unclear.

RESPONSE: Thank you for pointing this out. We have edited this text and hope it is clearer now. We now write:

“ Another way to look at overshoots is by considering the integrated overshoot temperature over time. When this overshoot integral is larger than 80 °C·years (Fig. 4b), the difference in year-2300 sea level rise between net-zero GHG and net-zero CO2 scenarios disappears. This equivalence can be understood better by looking at the characteristic temperature evolution in both scenario subsets (Fig. 1). Net-zero CO2 scenarios consistent with the Paris Agreement exhibit small but persistent overshoot, while net-zero GHG scenarios show higher but shorter overshoots. At an integral overshoot value larger than 80 °C·years, the additional early sea level rise pulse during the high overshoot in net-zero GHG scenarios cannot be compensated through lower rates of sea level rise later due to declining temperatures until 2300. Only a limited amount of sea level rise can be avoided through falling temperatures. ”

Line 133-135: Assuming the authors (or editors) take my advice and go back to a much more complete uncertainty analysis, the difference between MAGICC and GCM projections would be worth estimating as well.

RESPONSE:

We thank the reviewer for this advice. MAGICC drives sea level rise through global mean temperature in our model setup. The skill of emulating the global mean temperature response including its uncertainties has been shown in a number of studies (Meinshausen et al. 2009; Meinshausen, Raper, and Wigley 2011). The MAGICC response to extreme mitigation scenarios as presented here is less well explored as already noted in the paragraph the reviewer refers to. To provide more clarity to the reader on the uncertainty surrounding the sea level response to global mean temperature change,

we now compare our estimates of thermal expansion, which provide guidance on the quality of our combined model response, with GCM and EMIC estimates for the strong mitigation scenario RCP2.6. Table S3 provides our thermal expansion estimates for the years 2200 and 2300. We set these numbers in context to results from more complex models in the supplementary text:

“Thermal expansion is closely linked to the aggregated heat uptake of the ocean. We estimate thermal expansion with the constrained-extrapolation approach (Mengel et al. 2016), driven by the MAGICC (Meinshausen, Raper, and Wigley 2011; Meinshausen et al. 2009) global mean temperature evolution. The comparison of our thermal expansion estimate for the year 2300 for the RCP2.6 scenario (median: 31.1cm; central 66th percentile range: 20.0cm-45.7cm, relative to 2000, Table S3) can inform on the model performance. Other studies have reported 2300 steric sea level rise of 6-37cm relative to 1986-2005 for an ensemble of models of intermediate complexity (Zickfeld et al. 2013) and 14-27cm relative to 2006 for a set of 6 CMIP5 models (Yin 2012)). Sea level rise through thermal expansion is thus slightly higher for RCP2.6 than in more complex models, but of comparable range. In line with CMIP5 model experiments (Bouttes et al. 2013), our steric sea-level rise component does not exhibit a decline in sea level under the net-zero GHG scenarios until 2300, despite the projected GMT decline. Median sea level rise ranges from 30 to 38cm in 2300 for net-zero CO2 scenarios and 20 to 30cm for net-zero GHG scenarios (Table S4a and S4b).“

Line 139-141: this section is totally misleading for the reasons above. The authors try to imply that the outcomes for ice dynamics and other contributors are somehow well anchored before admitting later on that at least the ice dynamic term is not. This compounds the failure to mention the problem at the outset.

RESPONSE: We agree with the reviewer that in view of the new science concerning ice dynamics these lines were misleading and we thank the reviewer for pointing this out. We did not intend to claim that our estimates are consistent with the ice-sheet long-term contribution to sea level rise for all levels of warming. As now discussed in detail, atmosphere-driven ice sheet decay is of minor relevance for the scenarios considered, and potential warming-independent West Antarctic ice sheet collapse does not supersede our projections on the time frame considered. Indeed, the reviewer’s comment helped us to ensure that our results are presented in the right context so that no arbitrary comparisons are carried out between sea level rise estimates from the literature for very high emission scenarios and our estimates for very stringent mitigation scenarios. Besides the extended discussion, we now highlight the issue at the outset, see the last paragraph of the introduction:

“Our approach accounts for climate-driven dynamical Antarctic ice discharge but does not cover potential ice instabilities (Mouginot, Rignot, and Scheuchl 2014; Favier et al. 2014; Joughin, Smith, and Medley 2014; Pollard, DeConto, and Alley 2015). Such instabilities can significantly increase the Antarctic ice sheet contribution for scenarios with medium to strong warming, but their contributions are estimated to be small for the scenarios and the time frame we consider here (DeConto and Pollard 2016).“

Line 164-66: here we finally have a statement that encapsulates the problem.

RESPONSE: We agree with the reviewer that our initial submission did not provide the required context and information for the reader to understand what the potential influence of these contributions could be. We have now discussed this in detail and show that when comparing like with like (that is, the estimated contribution for the RCP2.6 scenario) our results are similar to the latest process-based estimates of DeConto and Pollard (2016). The upper one-sigma estimate (68th percentile) is 10cm to 20cm higher in the DeConto and Pollard (2016) study than in our estimate, we thus note that we do not fully encompass the DeConto and Pollard (2016) uncertainty.

Figures: what does “peaky” mean? Peak year? Where does upward dark blue triangle appear on the figures? Why no legend explaining solid line colors in Figure 2?

RESPONSE: Thank you for highlighting these issues. We changed the labeling to “peak year” in all figures and added more markers so the upward dark blue triangle becomes visible in Fig. 1. We added a legend to Figure 2.

Reviewer #2 (Remarks to the Author):

Review of: The sea level legacy of the Paris Agreement and the effect of delayed mitigation action
Submitted to Nature Communications by Mengel et al.

Overall this is a solid contribution, but I don't believe it elevates to a level worthy of publication in Nature Communications. The paper certainly does represent an incremental contribution to an important and hot topic—emissions associated with Paris temperatures pledges, and their implications for long term sea level rise given all the research community has been learning about irreversibility and inertia of the ice sheets especially. Unfortunately the paper only mentions all the recent advances in our understanding of ice sheets and irreversibility; the actual modeling and sea level rise projections rely on techniques that are not particularly fresh. The results similarly are straightforward extensions of what is already known.

In terms of the paper's advances, albeit incremental, to my knowledge the discussion of how a delay in emissions peak year impacts 2300 sea level rise is previously unreported (in the context of Paris targets), as is the discussion of how much CO₂ overshoot (in the context of Paris targets) impacts 2300 sea level rise. However, the answer to the former is pretty unsurprising (a 15 year delay might mean approximately an additional foot of total sea level rise by 2300) and the latter is barely discussed in the body of the manuscript. More fundamentally, it is becoming clearer that there are real limitations in the modeling frameworks the authors use, such as MAGICC and the IPCC AR5 sea level rise projections. While the authors do an adequate job of mentioning that these limitations exist, from my perspective a Nature Communications paper would need to a) actually incorporate into their analysis the latest understanding of how ice sheets may change (e.g Deconto and Pollard 2016, Bamber and Aspinall 2013 and a bevy of other post AR5-cutoff date papers) or b) weigh in on why their approach—based on a simplified set of assumptions—remains justified despite all the emerging science.

RESPONSE:

Thank you very much for this critical review of our manuscript. We fully agree with the reviewer that our initial manuscript did a poor job in setting our results in context concerning recent advances in our understanding of ice sheet dynamics. Indeed, the manuscript only mentioned these caveats at the very end, with little to no quantification of their potential influence. We equally agree that this recently emerged literature deserves a more detailed discussion in our manuscript, a discussion which can help the reader understand how these new insights would (or would not) affect our sea level rise estimates. To this end, we have reworked our discussion sections, which sets our estimates for the ice sheet contributions in a broader context. We now provide a detailed discussion of the responses of the sea level components, including numbers for the RCP2.6 scenario until 2300, in the supplementary information. We compare to published results and give a summary in the discussion section of the main text. We made this split to keep the manuscript concise, but are willing to merge the supplementary text into the main text if the reviewer finds this necessary.

Contributions of dynamic Antarctic ice sheet loss can be very large, as illustrated by many studies, including the ones cited by the reviewer. Irreversible ice sheet loss from West Antarctica, which may already be triggered today, is however improbable to fully unfold in the timescale assessed here. We now discuss this aspect in our manuscript. Because this contribution is largely independent of future atmospheric forcing, it adds to long-term sea level uncertainty (mainly beyond 2300), but does not disqualify our comparisons between climate-driven sea level rise. DeConto and Pollard (2016) suggest that atmospheric-driven strong increase in future ice discharge will dominate sea level rise under medium or high-emissions (and hence medium or high-temperature) futures. These are very different

from the stringent mitigation futures that fall within the constraints set by the Paris Agreement. Deconto and Pollard (2016) also provide estimates for RCP2.6, a scenario which in terms of forcing and temperature outcome straddles the high end of Paris-consistent pathways. Based on this estimate, we can carry out a like-with-like comparison of Antarctic ice sheet contributions showing that our RCP2.6 estimate is close to Deconto and Pollard (2016). We are thus confident that our method is applicable for the strong-mitigation scenarios we explore in this manuscript, but should not be used for scenarios with stronger warming. We clarify this further in the main manuscript at the end of the introduction:

“Our approach accounts for climate-driven dynamical Antarctic ice discharge but does not cover potential ice instabilities (Rignot et al. 2014; Favier et al. 2014; Joughin, Smith, and Medley 2014; Pollard, DeConto, and Alley 2015). Such instabilities can significantly increase the Antarctic ice sheet contribution for scenarios with medium to strong warming, but their contributions are estimated to be small for the scenarios and the time frame we consider here (DeConto and Pollard 2016).”

and in the discussion:

“Our method does not cover processes that become independent of the climate forcing, such as ice instabilities. Marine ice-sheet instability may be already under way (Rignot et al. 2014; Favier et al. 2014; Joughin, Smith, and Medley 2014), but its impact on sea level rise until year 2300 remains limited (Feldmann and Levermann 2015). Newly-proposed instabilities triggered by atmospheric warming (Pollard, DeConto, and Alley 2015) strongly increase Antarctica’s contribution for medium to strong warming scenarios (DeConto and Pollard 2016). For stringent mitigation scenarios similar to RCP2.6 as assessed in this study, these instabilities are unlikely to be triggered and our estimate of RCP2.6 is similar to published numbers including the potential contribution of these instabilities (DeConto and Pollard 2016). Importantly, our method can not be expanded beyond the 23rd century as sea level uncertainty through Antarctic ice instability will become more pronounced.”

More details are given in the new supplementary text:

“New literature foresees an increased risk for major Antarctic ice sheet loss. West Antarctic ice sheet collapse may already be underway, induced by warmer ocean waters (Rignot et al. 2014; Favier et al. 2014; Joughin, Smith, and Medley 2014; Feldmann and Levermann 2015). Such collapse, which can become independent of its trigger, is difficult to link to present anthropogenic climate change as well as future climate pathways. Even under the strongest applied forcing, a West Antarctic ice sheet collapse is not unfolding for centuries (Joughin, Smith, and Medley 2014). While (Joughin, Smith, and Medley 2014) do not provide rates of sea level rise for the main phase of the collapse, whole West Antarctica simulations indicate an upper bound of 5 cm in the first 200 years (Feldmann and Levermann 2015). The risk of ocean-driven marine-ice-sheet instability hence increases uncertainty in future sea level rise, but plays a minor role for the time scale considered here. Once triggered and independent of the forcing, it would not affect relative changes between scenarios.

In contrast to the ocean-induced collapse (Joughin, Smith, and Medley 2014; Feldmann and Levermann 2015), two newly-proposed instability mechanisms (Pollard, DeConto, and Alley 2015) suggest a tight link between future atmospheric warming and Antarctic ice discharge (DeConto and Pollard 2016). Such discharge thus depends on the emission scenario, with an estimated sea level contribution for medium and strong warming scenarios (RCP4.5 and RCP8.5) that is much higher than in earlier estimates (Church et al. 2013; Golledge et al. 2015; Mengel et al. 2016). These instability mechanisms, which are linked to atmospheric warming, are improbable to be triggered for the scenarios we assess here with similar or less global warming than RCP2.6. Our median 2300 sea level estimate for RCP2.6 for Antarctica (14.5cm, Table S3) is similar to (DeConto and Pollard 2016) (13.5 or 15.0 cm, depending on their Pliocene sea level target, linearly interpolated between the 2100 and

2500 values provided in the publication). Our method does however not fully incorporate the uncertainty related to these instabilities: our 1 sigma estimate (68th percentile) is 10cm and 20 cm below the (DeConto and Pollard 2016) estimate, depending on their Pliocene target. Importantly, our method can not be expanded beyond the 23rd century as sea level uncertainty through ice instability will become more pronounced. Median sea level rise from Antarctica ranges from 13.5 to 18cm in 2300 for our net-zero CO2 scenarios and 9 to 14 cm for net-zero GHG scenarios (Table S4o and S4p).”

Finally, we also appreciate the reviewer’s suggestion to more clearly highlight some of the key policy-relevant findings.

Again, I believe the paper with some minor work (see below) will deserve publication in a specialist journal, where it is common to make a set of simple assumptions based on models that have an established history such as MAGICC, and apply those models with a new angle (e.g. Paris targets, and 2300 projections). I think a Nature Communications paper would need to more deeply sample the true range of possible outcomes using this new angle (Paris targets, and 2300 projections). The absence of any real surprises in the results further undermines the case for publication in Nature Communications.

RESPONSE: We thank the reviewer for this assessment. As now explained in the discussion and the supplementary text, we are confident that our approach is valuable for the here-explored strong mitigation scenarios, but agree that it is not applicable to sample the true range of possible outcomes for higher emissions scenarios than RCP2.6. We now provide a detailed discussion for the sea level components in question and compare to the literature so that the reader can better understand the context of our results. We would furthermore like to highlight, that to our knowledge an assessment of 2300 SLR under Paris Agreement pathways has not been performed yet.

Below are minor comments that might help the authors should they seek publication in a specialist journal, where I am confident this paper will find a good home.

Line 34: ‘subsidy’ to ‘subsidence’ ?

RESPONSE: Sorry for this typo. We have revised the entire manuscript, including the Supplementary Information to avoid them in this revised version.

Line 35: Use a crisper reference here, such as Sweet and Park, 2014.

RESPONSE: We changed the reference to Sweet and Park, 2014

Line 43: Change citation to Jevrejeva et al.

RESPONSE: Sorry for the mistake. We removed the citation to Jevrejeva, Grinstead and Moore (2014) and keep Jevrejeva et al. 2016 and Kopp et al. 2016. Kopp et al. provide semi-empirical projections for the RCPs, see his Fig. 1e and 1f.

Lines 47-49: this is not a sentence

RESPONSE: Sorry. A verb was indeed missing. We corrected this.

Line 65-66: You have not yet said what the baseline sea level period is.

RESPONSE: Thank you. We clarified that all absolute sea level rise projections in our manuscript are expressed relative to 2000 levels.

Lines 78-79 and elsewhere: consider changing ‘until year 2300’ to ‘through at least the year 2300’ just to ensure no one thinks you are saying it changes in 2300

RESPONSE: We have made this clarification for all instances, at times using a different wording.

Line 115: There are grammar issues in the second half of this sentence.

RESPONSE: Thank you, we have edited this part of the result description.

Line 171: should read 'have yet'

RESPONSE: We rewrote this part of the discussion and do not use the expression any longer.

Other comments

The differences between, and assumptions being made about, net zero GHG vs. net zero CO₂ that is obliquely explained in lines 103-105 should be more directly explained at first use. It is hard to follow in the current version of the manuscript.

RESPONSE: We now introduce the two scenario subsets in our scenario ensemble at the beginning of the results section:

“We investigate scenarios that achieve the net-zero GHG emission goal of the Paris Agreement and hold global mean temperature rise at various levels below 2°C (Fig. 1d). We also explore scenarios that only achieve net-zero CO₂ emissions, while still stabilizing temperature rise below 2°C (Fig. 1a). Throughout the manuscript we discuss the results for these two subsets of our scenario ensemble.”

It strikes me as a bit awkward to have Figures 1a and 1b not called out until the 'end' of the paper (methods). Perhaps it is OK, depending on the journal; you might just think about pros and cons, and whether there is a better option.

RESPONSE: We agree that this was a bit awkward. We hope that referencing these two panels when also introducing our two scenario subsets (at the beginning of the results section) now avoids awkwardness and also gives a clearer understanding of our two scenario subsets.

The paper mentions a sufficient number of caveats, but for at least a couple of the biggies, you might add another sentence saying something about it. For example, it may be really important that the IPCC AR5 projections did not include Marine Ice Sheet Instability (even if we do not exceed 2C warming). As another example, is there an argument against trusting your Greenland reversal findings? At least a little discussion based the literature would seem to be justified here.

RESPONSE: Thank you very much. We fully agree, and have included a more detailed discussion of these issues as outlined above, in particular on Marine Ice Sheet Instability. We also added a discussion of the Greenland contribution in more detail. We included a figure on the rate of the contribution for each component (Fig. S5 for the net-zero GHG scenarios). With this additional information, we refined the text and now write:

“Under the Paris Agreement's net-zero GHG constraint and the implied declining global mean temperatures, the contributions from thermal expansion, mountain glaciers and Greenland surface mass balance can stabilize during the 22nd century (Fig. S3 and S5). Mountain glaciers may even contribute slightly negatively towards the end of the 22nd century in our model for the earliest peak and fastest reduction scenarios.”

Surface-mass-balance-driven irreversible Greenland ice sheet loss is a minor factor on the time scales and for the warming levels considered in this publication. We have added the following sentence to the discussion to clarify this:

“Surface-mass-balance-related instability of the Greenland ice sheet would evolve extremely slow under the assessed levels of global warming (Robinson, Calov, and Ganopolski 2012) and is thus of minor importance here.”

We also added a part to the supplementary text that critically discusses Greenland’s solid ice discharge since it becomes an important contributor in our model. This is referenced in the main-text discussion:

“Our estimate of Greenland’s solid ice discharge is above the range of ice-sheet model simulations, but these simulations do not fully resolve outlet glaciers dynamics and do not reflect the observed continuing and increasing solid ice discharge (Broeke et al. 2016).”

References

Bamber, J. L., and W. P. Aspinall (2013), An expert judgement assessment of future sea level rise from the ice sheets, *Nat. Clim. Change*, 3, 424–427, doi:10.1038/nclimate1778.

Sweet WV, Park J. From the extreme to the mean: Acceleration and tipping points of coastal inundation from sea level rise. *Earth's Future*. 2014;2(12):579-600. doi: 10.1002/2014EF000272.

Reviewer #3 (Remarks to the Author):

Overall this is an excellent and well-written paper. It addresses the issue of sea level commitment. The results of the paper will be of wide interest to a range of academics, policy makers and coastal stakeholders. I recommend publication if the following moderate-issues below are addressed:

RESPONSE: Thank you very much for the constructive review. We respond to all issues raised in our point-by-point response below.

1. I think the final paragraph introducing the methods on line 51-58 needs to be expanded. Currently it is too brief. I think you need to add a sentence or two outlining what scenarios you have run. On Line 67 you say 'for the three scenarios' but I am not sure until the methods section what these scenarios are! Please include a briefly description in the methods paragraph above regarding this.

RESPONSE: Thank you to highlight this need for clarification. The format guidelines keep us from including a too large part of the methods in the main text, but we have included a slightly longer description at the beginning of the results section. Moreover, we have also made sure to make any references to specific scenarios clearer throughout the text.

2. This is just a suggestion, but I recommend showing a sub-plot on Figure 1, beneath panels c and f, which show the rate of sea level rise per year. This will make it easier to see when things start flattening off.

RESPONSE: This is a very valuable suggestion. To keep Fig. 1 concise, we decided to create an extra figure. We show the rates of sea level rise in Fig. S2.

3. I think it would be good to stress in either the introduction or conclusion section the importance of considering the commitment of sea level rise in coastal planning. For example major schemes, such as the Thames Barrier, or the building of Nuclear power stations, with a life-time of more than 100 years, must account for sea level commitments beyond 2100.

RESPONSE: Thank you for this suggestion. We deliberately designed the paper to cover the extended time span until 2300, so that it is useful for coastal planning horizons that reach past 2100.

4. In the discussion the results of this study are not compared/contrasted with other efforts that have projected sea level out beyond 2100. I think a paragraph should be added describing briefly how the authors results agree with other published efforts, particularly from GCMs, to project sea level rise out beyond 2100.

RESPONSE: Thanks, we very much agree with this suggestion and have expanded the component-wise analysis of 2300 sea level rise. We have included RCP2.6 projections with our simplified approach that allow us to compare our findings with results from GCMs and EMICs until 2300. See also the new Figures S1 and S2 as well as Table S3 and S4 for comparison with RCP2.6. We find that our results for steric SLR compare well to projections from more complex models, although they are skewed towards the upper end of such model estimates.

We have included the following paragraph in the new supplementary text:

“Thermal expansion is closely linked to the aggregated heat uptake of the ocean. We estimate thermal expansion with the constrained-extrapolation approach (Mengel et al. 2016), driven by the MAGICC (Meinshausen, Raper, and Wigley 2011; Meinshausen et al. 2009) global mean temperature evolution. The comparison of our thermal expansion estimate for the year 2300 for the RCP2.6 scenario (median: 31.1cm; central 66th percentile range: 20.0cm-45.7cm, relative to 2000, Table S3) can inform on the model performance. Other studies have reported 2300 steric sea level rise of 6-

37cm relative to 1986-2005 for an ensemble of models of intermediate complexity (Zickfeld et al. 2013) and 14-27cm relative to 2006 for a set of six CMIP5 models (Yin 2012). Sea level rise through thermal expansion is thus slightly higher for RCP2.6 than in more complex models, but of comparable range. In line with CMIP5 model experiments (Bouttes et al. 2013), our steric sea-level rise component does not exhibit a decline in sea level under the net-zero GHG scenarios until 2300, despite the projected GMT decline. Median sea level rise ranges from 30 to 38cm in 2300 for net-zero CO₂ scenarios and 20 to 30cm for net-zero GHG scenarios (Table S4a and S4b).”

5. The method section in my opinion is way to brief. Although the model used to project future sea level rise is described in another paper (Mengel et al., 2016) I still think it is important here in this current paper, that it is at least described in some way. Could I encourage the authors to add another paragraph briefly describing how the contribution-based semi-empirical model works.

RESPONSE: Following the referee’s suggestion, we have expanded the Methods section. We now provide a dedicated paragraph “global sea level projections”, which describes the Mengel et al. (2016) model in greater detail.

References

- Bouttes, N., J. M. Gregory, J. A. Lowe, N. Bouttes, J. M. Gregory, and J. A. Lowe. 2013. "The Reversibility of Sea Level Rise." Research-article. [Http://Dx.Doi.Org/10.1175/JCLI-D-12-00285.1](http://Dx.Doi.Org/10.1175/JCLI-D-12-00285.1). April 23. <http://journals.ametsoc.org/doi/abs/10.1175/JCLI-D-12-00285.1>.
- Broeke, Michiel R. van den, Ellyn M. Enderlin, Ian M. Howat, Peter Kuipers Munneke, Brice P. Y. Noël, Willem Jan van de Berg, Erik van Meijgaard, and Bert Wouters. 2016. "On the Recent Contribution of the Greenland Ice Sheet to Sea Level Change." *The Cryosphere* 10 (5): 1933–46. doi:10.5194/tc-10-1933-2016.
- Church, J. A., P.U. Clark, A. Cazenave, J.M. Gregory, S. Jevrejeva, A. Levermann, M.A. Merrifield, et al. 2013. "Chapter 13. Sea Level Change." In *Climate Change 2013: The Physical Science Basis. Contribution of Working Group I to the Fifth Assessment Report of the Intergovernmental Panel on Climate Change*, edited by T. F. Stocker, D. Qin, G.-K. Plattner, M. Tignor, S.K. Allen, J. Boschung, A. Nauels, Y. Xia, V. Bex, and P. M. Midgley. Cambridge University Press, Cambridge, United Kingdom and New York, NY, USA.
- DeConto, Robert M., and David Pollard. 2016. "Contribution of Antarctica to Past and Future Sea-Level Rise." *Nature* 531 (7596): 591–97. doi:10.1038/nature17145.
- Favier, L., G. Durand, S. L. Cornford, G. H. Gudmundsson, O. Gagliardini, F. Gillet-Chaulet, T. Zwinger, A. J. Payne, and A. M. Le Brocq. 2014. "Retreat of Pine Island Glacier Controlled by Marine Ice-Sheet Instability." *Nature Climate Change* 4 (2): 117–21. doi:10.1038/nclimate2094.
- Feldmann, Johannes, and Anders Levermann. 2015. "Collapse of the West Antarctic Ice Sheet after Local Destabilization of the Amundsen Basin." *Proceedings of the National Academy of Sciences* 112 (46): 14191–96. doi:10.1073/pnas.1512482112.
- Golledge, N. R., D. E. Kowalewski, T. R. Naish, R. H. Levy, C. J. Fogwill, and E. G. W. Gasson. 2015. "The Multi-Millennial Antarctic Commitment to Future Sea-Level Rise." *Nature* 526 (7573): 421–25. doi:10.1038/nature15706.
- Joughin, I., B. E. Smith, and B. Medley. 2014. "Marine Ice Sheet Collapse Potentially Under Way for the Thwaites Glacier Basin, West Antarctica." *Science* 344 (6185): 735–38. doi:10.1126/science.1249055.
- Meinshausen, Malte, Nicolai Meinshausen, William Hare, Sarah C. B. Raper, Katja Frieler, Reto Knutti, David J. Frame, and Myles R. Allen. 2009. "Greenhouse-Gas Emission Targets for Limiting Global Warming to 2 °C." *Nature* 458 (7242): 1158–62. doi:10.1038/nature08017.
- Meinshausen, Malte, S. C. B. Raper, and T. M. L. Wigley. 2011. "Emulating Coupled Atmosphere-Ocean and Carbon Cycle Models with a Simpler Model, MAGICC6 – Part 1: Model Description and Calibration." *Atmospheric Chemistry and Physics* 11 (4): 1417–56. doi:10.5194/acp-11-1417-2011.
- Mengel, Matthias, Anders Levermann, Katja Frieler, Alexander Robinson, Ben Marzeion, and Ricarda Winkelmann. 2016. "Future Sea Level Rise Constrained by Observations and Long-Term Commitment." *Proceedings of the National Academy of Sciences* 113 (10): 2597–2602. doi:10.1073/pnas.1500515113.
- Mouginot, J., E. Rignot, and B. Scheuchl. 2014. "Sustained Increase in Ice Discharge from the Amundsen Sea Embayment, West Antarctica, from 1973 to 2013." *Geophysical Research Letters* 41 (5): 1576–84. doi:10.1002/2013GL059069.
- Pollard, David, Robert M. DeConto, and Richard B. Alley. 2015. "Potential Antarctic Ice Sheet Retreat Driven by Hydrofracturing and Ice Cliff Failure." *Earth and Planetary Science Letters* 412 (February): 112–21. doi:10.1016/j.epsl.2014.12.035.
- Robinson, Alexander, Reinhard Calov, and Andrey Ganopolski. 2012. "Multistability and Critical Thresholds of the Greenland Ice Sheet." *Nature Climate Change* 2 (6): 429–32. doi:10.1038/nclimate1449.
- Yin, Jianjun. 2012. "Century to Multi-Century Sea Level Rise Projections from CMIP5 Models." *Geophysical Research Letters* 39 (17): L17709. doi:10.1029/2012GL052947.

Zickfeld, Kirsten, Michael Eby, Andrew J. Weaver, Kaitlin Alexander, Elisabeth Crespin, Neil R. Edwards, Alexey V. Eliseev, et al. 2013. "Long-Term Climate Change Commitment and Reversibility: An EMIC Intercomparison." *Journal of Climate* 26 (16): 5782–5809. doi:10.1175/JCLI-D-12-00584.1.

Reviewers' comments:

Reviewer #1 (Remarks to the Author):

The authors have responded comprehensively to my earlier criticisms. Overall, the manuscript is now in good shape with one minor exception: I find the abstract's fourth and fifth sentences related to emissions scenario assumptions to be unclear and suggest the authors re-read and rephrase these, perhaps showing a revised version of the abstract to someone outside the author team to assure clarity.

Reviewer #2 (Remarks to the Author):

I find the reviewed manuscript to be somewhat stronger. For example, the grammatical errors have been dealt with, the key concepts are now better explained when they are introduced (e.g. the net zero GHG and CO₂ distinctions), and the caveats are less buried at the end of the paper. These improvements to my mind make the paper even more appropriate than before for a specialist journal.

I simply do not see the case for publishing this paper in Nature Communications, however. The fundamentals of this paper have not changed—it has been cleaned up and polished. This is fundamentally an incremental advance only. The modeling framework is not state of the art by any means, and I find the treatment of tail risks to be facile and superficial, in light of all the lines of evidence pointing to the non-negligible risk of rapid sea level change.

Fundamentally, the authors seem to be stating that while their modeling approach is far from state of the art, it is sufficient because you only need to include cutting edge ice sheet science, modeling and perspectives that integrate multiple lines of evidence and expert judgment if considering RCPs above 2.6 and years beyond 2300. To my mind these statements are not sufficiently supported by the authors. More specifically:

The authors are far too confident that 2.6 won't lead to large SLR by 2300. Their primary argument here seems to be that Deconto and Pollard did not see significant ice sheet instability and additional sea level rise under RCP 2.6 and 2300 in their model. But Deconto and Pollard is one study (albeit an important one) focused on one potential instability mechanism, in Antarctica only. I believe a Nature Communications paper should be holistic and critical, rather than leaning on one influential study. Even if we accepted the authors' premise that RCP 2.6 won't lead to instability in the near future, the authors' assumption that it won't do so before 2300 strikes me as too breezy and glib for Nature Communications.

While the authors do mention Greenland, the treatment is even more superficial than for Antarctica. The reader is essentially told that Greenland won't produce large melt under RCP2.6. Again, in a specialist journal, applying an older model to a rapidly evolving and cutting edge problem (sea level rise) might be OK, if the authors briefly acknowledged all the limitations of their modeling framework (which they generally do, although not with depth [e.g. discussion of the universe of processes and mechanisms that can impact ice sheets]). But a Nature Communications piece, to my mind should not dismiss the key uncertainties in such a facile way, essentially saying that a couple (out of a broader universe) of the potential surprises in the system are 'unlikely' to be relevant for RCP 2.6 and 2300. I find the paper to push to the side precisely the aspects that from a probabilistic perspective can have the largest impact on how much sea level rise can be experienced by 2300 (even under RCP2.6). What they do report on has one innovation—it speaks to Paris and RCP2.6 out to 2300—but it is largely an accounting exercise since it deals only with the tidy elements of sea level rise and ignores the key sources of uncertainty. (As an aside, even within the tidy elements [e.g. thermal expansion] I would have expected a Nature Communications piece to speak to whether there is any evidence that has emerged in the 5+

years suggesting GCM scaling/MAGICC type approaches could miss certain tail outcomes]. Again, the paper is perfectly fine for a specialist journal. But in Nature Communications, I do not feel the cutting edge science can be kept out of the analysis simply because the authors deem it 'unlikely' based on a superficial review of the literature in this burgeoning field.

Reviewer #3 (Remarks to the Author):

The authors have done a good job responding to mine and the other reviewers comments. The paper has significantly improved as a results. I recommend the paper is accepted. I have a minor comment. I wonder whether the first sentence would read better if the work impact was replaced with consequence, ie. 'Sea level rise is one of the major consequences of climate change'.

Response to the reviewers for manuscript

“The sea-level legacy of the Paris Agreement and the effect of delayed mitigation action” with manuscript ID NCOMMS-17-00342A.

Reviewers' comments:

Reviewer #1 (Remarks to the Author):

The authors have responded comprehensively to my earlier criticisms. Overall, the manuscript is now in good shape with one minor exception: I find the abstract's fourth and fifth sentences related to emissions scenario assumptions to be unclear and suggest the authors re-read and rephrase these, perhaps showing a revised version of the abstract to someone outside the author team to assure clarity.

Response:

We thank the reviewer for re-reading the manuscript and her/his positive assessment. We discussed the abstract with colleagues from other disciplinary fields and rewrote it to improve clarity.

Reviewer #2 (Remarks to the Author):

I find the reviewed manuscript to be somewhat stronger. For example, the grammatical errors have been dealt with, the key concepts are now better explained when they are introduced (e.g. the net zero GHG and CO₂ distinctions), and the caveats are less buried at the end of the paper. These improvements to my mind make the paper even more appropriate than before for a specialist journal.

Response:

We thank the reviewer for acknowledging the improvements we implemented during the first review round of our manuscript.

I simply do not see the case for publishing this paper in Nature Communications, however. The fundamentals of this paper have not changed—it has been cleaned up and polished. This is fundamentally an incremental advance only. The modeling framework is not state of the art by any means, and I find the treatment of tail risks to be facile and superficial, in light of all the lines of evidence pointing to the non-negligible risk of rapid sea level change.

Response:

We appreciate the reviewer's point that our manuscript was not strong on including tail risks that can emerge even under scenarios of strong climate mitigation. However, regarding the modeling framework, we do not agree with the reviewer's strong judgement. Except for advances in our understanding of the Antarctic ice sheet response, we are not aware of scientific leaps that render our approach (Mengel et al. 2016) outdated for the application in the context of a global multi-scenario assessment. We acknowledge that our treatment of the Antarctic ice sheet response was a weakness in our earlier submission. We thank the reviewer for his/her emphasis on the importance of the new insights, and

now include a new approach for Antarctica, which also covers the potential for rapid sea level change. Though median sea level estimates do not change much, the new approach affects our upper end sea level estimates, mainly due to the higher numbers for Antarctica. Moreover, our recalibration of the Greenland component with newer data also adds to the higher upper end estimates. These adjustments indeed significantly expand the uncertainty space.

We thus are thankful that the reviewer suggested these improvements, which gave us the chance to revise the high-risk low-probability estimates under strong mitigation scenarios. Details of the new Antarctic methodology are discussed in the Supplementary information, see also (Nauels et al. 2017), where we firstly applied the parametrization within a different modeling framework.

Fundamentally, the authors seem to be stating that while their modeling approach is far from state of the art, it is sufficient because you only need to include cutting edge ice sheet science, modeling and perspectives that integrate multiple lines of evidence and expert judgment if considering RCPs above 2.6 and years beyond 2300. To my mind these statements are not sufficiently supported by the authors.

Response:

For Antarctica, we agree with the author that an update was needed to be consistent with recent scientific advances to ensure credibility. Hence, we revised our methodology with a new Antarctic ice sheet parametrization to cover newly proposed ice-loss mechanisms. While other methodologies exist, for example incorporating expert elicitation, the presented parametrizations keep a strong link to physical modelling, using easily interpretable numbers like response timescales or threshold temperatures. We provide a reasoning for the Greenland ice sheet further below.

More specifically:

The authors are far too confident that 2.6 won't lead to large SLR by 2300. Their primary argument here seems to be that Deconto and Pollard did not see significant ice sheet instability and additional sea level rise under RCP 2.6 and 2300 in their model. But Deconto and Pollard is one study (albeit an important one) focused on one potential instability mechanism, in Antarctica only. I believe a Nature Communications paper should be holistic and critical, rather than leaning on one influential study. Even if we accepted the authors' premise that RCP 2.6 won't lead to instability in the near future, the authors' assumption that it won't do so before 2300 strikes me as too breezy and glib for Nature Communications.

Response:

We updated our approach for Antarctica, which now includes rapid ice loss. The Antarctic ice sheet is more sensitive to strong warming in the updated approach. Our climate response ensemble includes some members with strong warming even under the strong mitigation scenarios here. The probabilistic combination of climatic and Antarctic response in our updated approach therefore results in a tail risk of high sea level rise under the assessed scenarios. We were not fully aware of this tail risk and we thank the reviewer to insist on these improvements. As we are not aware of any comparable studies, we rely on the Deconto and Pollard (Nature 2016) study for our parametrization. There is a number of studies that propose marine ice sheet instability in the Amundsen region is already

under way or possible in the near future (Joughin, Smith, and Medley 2014; Mouginot, Rignot, and Scheuchl 2014; Favier et al. 2014). Though called instability, such ice sheet retreat occurs on centennial or slower timescales. The rates of such retreat would be within the range of our updated estimates. For example, Feldmann and Levermann (2015) estimate a contribution of 7.5 cm until 2300 through West Antarctic ice sheet collapse in ice-dynamical contributions. This estimate may be corrected upwards, but indicates that a multi-meter contribution through marine ice sheet instability alone from West Antarctica is not probable until 2300.

While the authors do mention Greenland, the treatment is even more superficial than for Antarctica. The reader is essentially told that Greenland won't produce large melt under RCP2.6. Again, in a specialist journal, applying an older model to a rapidly evolving and cutting edge problem (sea level rise) might be OK, if the authors briefly acknowledged all the limitations of their modeling framework (which they generally do, although not with depth [e.g. discussion of the universe of processes and mechanisms that can impact ice sheets]). But a Nature Communications piece, to my mind should not dismiss the key uncertainties in such a facile way, essentially saying that a couple (out of a broader universe) of the potential surprises in the system are 'unlikely' to be relevant for RCP 2.6 and 2300. I find the paper to push to the side precisely the aspects that from a probabilistic perspective can have the largest impact on how much sea level rise can be experienced by 2300 (even under RCP2.6).

Response:

We agree that some processes of the Greenland ice sheet that recently emerged in the literature are not covered by our approach and their discussion deserved more attention in the manuscript. This is in particular true for the feedback between surface melting and the radiation budget. To our knowledge, modeling on the involved processes is not mature enough so that they could be incorporated in parameterizations like ours. We now provide more details on these caveats in the discussion and the Supplementary Information. We comprehensively covered the relevant literature we were able to identify through literature searches as no specific references were provided by the referee. Furthermore, we recalibrated our approach with new observations (Broeke et al. 2016; Forsberg, Sørensen, and Simonsen 2017). We now display the total contribution from Greenland in Fig. 2 to make it visually comparable to the Antarctic contribution. Numbers for the individual processes are provided in the Supplementary Information as before. The Supplementary Information also includes a description of the updated calibration and our new calibration parameters.

The discussion on Greenland in the main text reads:

"For the Greenland ice sheet, potential self-sustained dynamics are not included in our parametrization though feedbacks exist and their future role for Greenland ice loss is not fully clear. The melt-elevation feedback - lower elevation leads to increased melt, which lowers elevation - can lead to threshold behaviour and long-term decay of the ice sheet, but it would evolve slowly under the levels of warming assessed here. The melt-albedo feedback - melting exposes darker ice, which absorbs more heat and increases melt - is not yet incorporated fully in process-based simulations. It is also not explicitly modelled in our approach and enters only indirectly through the calibration to recent observations of high surface melting. Anomalous atmospheric conditions are proposed as the main cause for this recent high melting. This makes it less probable that future ice loss is dominated by the melt elevation feedback. Our Greenland solid ice discharge parametrization does

not account for ice flow instability. The potential for large-scale self-accelerating marine-ice-sheet-instability-driven ice loss like in Antarctica is limited in Greenland as it does not have large marine ice basins that are open to the ocean. Warmer ocean temperatures will still affect ice loss, which is covered in our approach. Our estimate of Greenland's solid ice discharge is above the range of ice-sheet model simulations, but these simulations do not fully resolve outlet glaciers dynamics and do not fully reflect the observed continuing solid ice discharge. Nevertheless, the absence of feedbacks in the Greenland mass balance representation of our approach is a caveat and we cannot rule out that such feedbacks add to the presented numbers here. More details are given in the Supplementary Information."

The longer discussion on Greenland in the Supplementary information reads:

"For Greenland, the marine-grounded regions that are open towards the ocean are limited, ice drains predominantly through narrow fjords. The potential for large-scale self-accelerating marine ice sheet instability hence is also limited, in contrast to the Antarctic ice sheet. Warmer ocean temperatures will still affect ice loss, but this contribution is covered in our response-function approach. The approach yields continuing Greenland solid ice discharge throughout 2300 (median for RCP2.6 scenario: 23.5 cm, Table S3b). This differs from earlier observations of reduced discharge (Enderlin et al. 2014) and ice sheet simulations showing a diminishing contribution from solid ice discharge (Fürst, Goelzer, and Huybrechts 2015; Vizcaino et al. 2015; Peano et al. 2017). These process-based simulations can however not fully resolve the narrow outlet glacier flow to the ocean that is central for solid ice discharge. They do not incorporate the deeper and larger subglacial basins (Morlighem et al. 2017) and deeper fjords (Rignot et al. 2016) in new ice and ocean floor data, which suggest increased sensitivity to climate warming. While new processes may emerge that are not covered here, we see our approach as a valid physical approximation of the discharge's ongoing response to future warming. The broad range (16-92 cm in 2300 for the RCP2.6 scenario, Table S3b) reflects at least partly our incomplete knowledge of this term.

Two feedbacks related to the surface mass balance may foster a self-sustained decay of the Greenland ice sheet. The melt-elevation feedback (Robinson, Calov, and Ganopolski 2012) can add to the directly temperature-driven sea-level contribution reflected in our methods. In the model of (Robinson, Calov, and Ganopolski 2012) the decay of the ice sheet occurs on a time scale longer than 10.000 years for 2°C temperature increase over Greenland (their Fig. 3). In the SRES A1B scenario (Nakicenovic et al. 2000), a scenario without any climate policy and estimated year-2100 warming of 3 to 4°C (Rogelj, Meinshausen, and Knutti 2012), the ice loss through this feedback has been estimated to be between 3.6 and 16% of the forcing-driven ice loss in 2200 (Edwards et al. 2014). Relating this fraction to our scenarios with lower warming is however not straightforward. Still, both Robinson et al. (2012) and Edwards et al. (2014) show the bounded role of the feedback. We thus do not see the melt-elevation feedback sufficiently large to dismiss or invalidate our approach on the time scales and warming levels assessed here. In two recent publications process-based ice sheet models have been used to project future Greenland ice loss. (Fürst, Goelzer, and Huybrechts 2015) find a total (surface-mass-balance and ice-dynamic) contribution of 9 cm under RCP2.6 until 2300. Peano et al. (2017) find 10 cm surface mass balance contribution from Greenland until 2100 under the RCP4.5 scenario. Fettweis et al. (2013) find 4 (2-6) cm for the same scenario until 2100. The melt-albedo feedback (Tedesco et al. 2016) is not fully integrated in these studies and may render these model results too low. This feedback has a physical (less snow and more bare ice absorb more radiation) and a biological component (wetter conditions allow more ice surface algae growth (Stibal et al. 2017; Tedstone et al. 2017)).

There are currently no process-based simulations available that estimate future Greenland ice loss while fully including this feedback. Tedesco et al. (2016) show that the extreme 2012 conditions had similar low ice albedo conditions as projected for the end of the century, and therewith highlighted that models are still incomplete for such projections. Assuming that yearly-repeated 2012 conditions can be used as an upper bound for climate scenarios that stay below 2°C, the upper bound of 670 Gt mass loss in 2012 would lead to 56 cm sea-level rise when summed up for 300 years. This is about 13cm above our 95th percentile surface mass balance estimate for Greenland for RCP2.6. If the melt-albedo feedback becomes a major driver of Greenland ice loss, these values could be exceeded. Research from this year however suggest that while the melt-albedo feedback enhances the ice loss, changes in the atmospheric circulation are the ultimate driver of the high melting in recent years (Hofer et al. 2017; Tedstone et al. 2017). A runaway feedback between atmospheric changes and the Greenland melt is not evident, which makes a self-sustained ice sheet collapse (as compared to climate-forcing driven collapse) less probable. Median sea-level rise from the combined Greenland surface mass balance and solid ice discharge ranges from 45 to 61 cm in 2300 for our net-zero CO2 scenarios and 32 to 48 cm for net-zero GHG scenarios (Table S4i and S4j). Values for the individual Greenland components are given in tables S4e to S4h.”

What they do report on has one innovation—it speaks to Paris and RCP2.6 out to 2300—but it is largely an accounting exercise since it deals only with the tidy elements of sea level rise and ignores the key sources of uncertainty. (As an aside, even within the tidy elements [e.g. thermal expansion] I would have expected a Nature Communications piece to speak to whether there is any evidence that has emerged in the 5+ years suggesting GCM scaling/MAGICC type approaches could miss certain tail outcomes]. Again, the paper is perfectly fine for a specialist journal. But in Nature Communications, I do not feel the cutting edge science can be kept out of the analysis simply because the authors deem it ‘unlikely’ based on a superficial review of the literature in this burgeoning field.

Response:

We thank the reviewer for this critical view. It motivated us to redo our Antarctic contribution as discussed in detail above. The updated methodology brings to the fore the risk for a low-probability, but high sea level rise contribution from Antarctica. We incorporated this important issue into the manuscript at several points. Concerning the Greenland contribution, we updated the calibration, expanded the discussion and chose more careful wording to better communicate that dynamics may emerge that would not be covered by our approach.

Reviewer #3 (Remarks to the Author):

The authors have done a good job responding to mine and the other reviewers comments. The paper has significantly improved as a results. I recommend the paper is accepted. I have a minor comment. I wonder whether the first sentence would read better if the work impact was replaced with consequence, ie. 'Sea level rise is one of the major consequences of climate change'.

Response:

We are pleased that the reviewer positively assesses our revisions. We replaced 'impacts' by 'consequence' in the first sentence of the abstract.

References

- Broeke, Michiel R. van Den, Ellyn M. Enderlin, Ian M. Howat, Peter Kuipers Munneke, Brice P. Y. Noël, Willem Jan van de Berg, Erik van Meijgaard, and Bert Wouters. 2016. "On the Recent Contribution of the Greenland Ice Sheet to Sea Level Change." *The Cryosphere* 10 (5). Copernicus GmbH: 1933–46.
- Favier, L., G. Durand, S. L. Cornford, G. H. Gudmundsson, O. Gagliardini, F. Gillet-Chaulet, T. Zwinger, A. J. Payne, and A. M. Le Brocq. 2014. "Retreat of Pine Island Glacier Controlled by Marine Ice-Sheet Instability." *Nature Climate Change* 4 (2). Nature Publishing Group: nclimate2094.
- Feldmann, J., and A. Levermann. 2015. "Collapse of the West Antarctic Ice Sheet after Local Destabilization of the Amundsen Basin." *Proceedings of the National Academy of Sciences of the United States of America* 112 (46): 14191–96.
- Forsberg, Rene, Louise Sørensen, and Sebastian Simonsen. 2017. "Greenland and Antarctica Ice Sheet Mass Changes and Effects on Global Sea Level." *Surveys in Geophysics* 38 (1). Springer Netherlands: 89–104.
- Joughin, Ian, Benjamin E. Smith, and Brooke Medley. 2014. "Marine Ice Sheet Collapse Potentially under Way for the Thwaites Glacier Basin, West Antarctica." *Science* 344 (6185): 735–38.
- Mengel, Matthias, Anders Levermann, Katja Frieler, Alexander Robinson, Ben Marzeion, and Ricarda Winkelmann. 2016. "Future Sea Level Rise Constrained by Observations and Long-Term Commitment." *Proceedings of the National Academy of Sciences of the United States of America* 113 (10): 2597–2602.
- Mouginot, J., E. Rignot, and B. Scheuchl. 2014. "Sustained Increase in Ice Discharge from the Amundsen Sea Embayment, West Antarctica, from 1973 to 2013." *Geophysical Research Letters* 41 (5): 1576–84.
- Nauels, Alexander, Joeri Rogelj, Carl-Friedrich Schleussner, Malte Meinshausen, and Matthias Mengel. 2017. "Linking Sea Level Rise and Socioeconomic Indicators under the Shared Socioeconomic Pathways." *Environmental Research Letters: ERL [Web Site]* 12 (11). IOP Publishing: 114002.

Reviewers' Comments:

Reviewer #2:

Remarks to the Author:

To my mind, the authors have now sufficiently addressed the potential limitations of their approach, thanks to a deeper dive into the latest literature on Greenland and Antarctic Ice Sheets. I also appreciate the minor modifications to their analyses. I recommend publication.